# Interpolation can hurt robust generalization even when there is no noise

**Konstantin Donhauser[1]\***, **Alexandru Țifrea[1]\***, **Michael Aerni[1]**,
**Reinhard Heckel[2,3]**, **Fanny Yang[1]**
[1]ETH Zurich   [2]Rice University   [3]Technical University of Munich
konstantin.donhauser@ai.ethz.ch tifreaa@inf.ethz.ch aernim@ethz.ch
reinhard.heckel@gmail.com fan.yang@inf.ethz.ch

## Abstract

Numerous recent works show that overparameterization implicitly reduces variance for min-norm interpolators and max-margin classifiers. These findings suggest that ridge regularization has vanishing benefits in high dimensions. We challenge this narrative by showing that, even in the absence of noise, avoiding interpolation through ridge regularization can significantly improve generalization. We prove this phenomenon for the robust risk of both linear regression and classification, and hence provide the first theoretical result on *robust overfitting*.

## 1   Introduction

Conventional statistical wisdom cautions the user who trains a model by minimizing a loss $\mathcal{L}(\theta)$: if a global minimizer achieves zero or near-zero training loss (i.e., it *interpolates*), we run the risk of overfitting (i.e., high variance) and thus suboptimal prediction performance. Instead, *regularization* is commonly used to reduce the effect of noise and to obtain an estimator with better generalization. Specifically, regularization limits model complexity and induces worse data fit, for example via an explicit penalty term $R(\theta)$. The resulting penalized loss $\mathcal{L}(\theta) + \lambda R(\theta)$ explicitly imposes certain structural properties on the minimizer. This classical rationale, however, does seemingly not apply to overparameterized models: for example, large neural networks in practice exhibit good generalization performance on i.i.d. samples even if $\mathcal{L}(\theta)$ vanishes and label noise is present [36].

Since interpolators are not unique in the overparameterized regime, it is crucial to study the specific *implicit biases* of interpolating estimators. In particular, for common losses, a large body of recent work analyzes the properties of the solutions found via gradient descent at convergence (see e.g. [44, 12, 13, 22, 26, 27, 49, 31]). For example, for linear and logistic regression, it is well-known that gradient descent converges to the min-$\ell_2$-norm and max-$\ell_2$-margin solutions, respectively [22, 27, 32, 49]. These interpolating estimators also minimize the respective penalized loss $\mathcal{L}(\theta) + \lambda\|\theta\|_2^2$ in the limit of $\lambda \to 0$ [44].

A plethora of recent papers explicitly study generalization properties on min-$\ell_2$-norm interpolators [15, 18, 23, 6, 33, 34, 35] and max-$\ell_2$-margin solutions [14, 34, 46], and show that the variance decreases as the overparameterization ratio increases beyond the interpolation threshold. While regularization with $\lambda > 0$ is commonly known to reduce the risk at the interpolation threshold [23, 37], many of these works are motivated by the second descent of the double descent phenomenon [7] which suggests that regularization becomes redundant with sufficient overparameterization. Hence, previous papers focus on highly overparameterized settings where the optimal regularization parameter satisfies $\lambda_{\text{opt}} \leq 0$ [29, 54, 43], implying that for large $d/n$, explicit regularization with $\lambda > 0$ is redundant or even detrimental for generalization.

---

*\*Equal contribution.*

35th Conference on Neural Information Processing Systems (NeurIPS 2021).

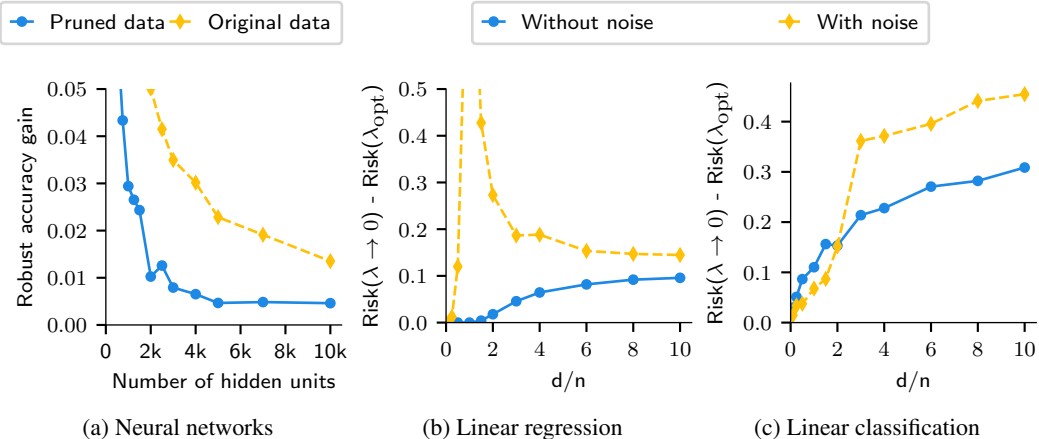

| | |
|---|---|
| (a) Neural networks | (b) Linear regression | (c) Linear classification |

Figure 1: Avoiding interpolation can benefit robustness even in the overparameterized ($d \gg n$) regime and for noiseless training data. We plot the robust accuracy gain of (a) early-stopped neural networks compared to models at convergence, fit on sanitized (binary 1-3) MNIST that arguably has minimal noise; and $\ell_2$ regularized estimators compared to interpolators with $\lambda \to 0$ for (b) linear regression with $n = 10^3$ and (c) robust logistic regression with $n = 10^3$. See Appendix B for experimental details and Sections 3 and 4 for the precise settings of (b) and (c).

Taking a step back, this narrative originated from theoretical and experimental findings that consider the *standard* test risk with identically distributed training and test data. However, this measure cannot reflect the *robust risk* of models when the test data has a shifted distribution, is attacked by adversaries, or contains many samples from minority groups [19, 21, 40, 55]. In fact, mounting empirical evidence suggests that regularization is indeed helpful for robust generalization, even in highly overparameterized regimes where the benefits for the standard risk are negligible [42]. This phenomenon is sometimes referred to as *robust overfitting*.

In the presence of noise, the following intuition holds true: since the robust risk amplifies estimation errors, its variance is larger and hence regularization – such as early stopping – can be beneficial for generalization [47]. However, we observe that even when estimating entirely noiseless signals, robust overfitting persists! We observe this phenomenon in experiments with shallow neural networks on sanitized image data depicted in Figure 1a and, in fact, even for linear models trained on high-dimensional synthetic noiseless data. In particular, Figures 1b,1c show that min-$\ell_2$-norm and robust max-$\ell_2$-margin interpolators (minimizers of the training loss for $\lambda \to 0$), achieve a higher robust risk than the corresponding regularized estimators that do not interpolate noiseless observations (minimizers for $\lambda > 0$).

To date, our observations in the noiseless case cannot be explained by prior work. On the contrary, they seem to contradict a simple intuition: if the min-$\ell_2$-norm and robust max-$\ell_2$-margin interpolators exhibit large risks as $\lambda \to 0$, the induced bias for a small $\ell_2$-norm is potentially suboptimal and a larger penalty weight $\lambda > 0$ should only degrade performance. In this paper, we provide possible explanations that debunk this intuition in the high-dimensional overparameterized regime. We prove for isotropic Gaussian covariates that a strictly positive regularization parameter $\lambda$ systematically improves robust generalization for linear and robust logistic regression. Empirically, we show that early stopping and other factors that lead to a non-interpolating estimator achieve a similar effect. Our results provide the first rigorous explanation of robust overfitting even in the absence of noise.

In Section 2, we formally define the setting that we use throughout our analysis. We then first present precise asymptotic expressions for the robust risk for linear regression in Section 3 that explicitly explain robust overfitting. Furthermore, in Section 4, we consider classification with logistic regression and derive asymptotic results.

## 2 Risk minimization framework

In this section, we introduce the data generating process that we assume throughout our analysis, and define the standard and robust risks that we use as evaluation metrics.

## 2.1 Problem setting

This paper considers the supervised learning problem of estimating a mapping from $d$-dimensional real-valued features $x \in \mathbb{R}^d$ to a target $y \in \mathcal{Y} \subseteq \mathbb{R}$ given a training set of labeled samples $\mathcal{D} = \{(x_i, y_i)\}_{i=1}^n$. We assume that the feature vectors $x_i$ are drawn i.i.d. from the marginal distribution $\mathbb{P}$ that we assume to be an isotropic Gaussian. We further focus on noiseless observations $y_i = \langle \theta^\star, x_i \rangle$ for regression tasks and $y_i = \text{sgn}(\langle \theta^\star, x_i \rangle)$ for classification tasks, respectively. However, the main results are more general and apply to noisy observations as well. For regression, we assume additive Gaussian noise with zero mean and $\sigma^2$ variance, while for classification we flip a certain percentage of the training labels.

This paper studies the high-dimensional asymptotic regime where $d/n \to \gamma$ as both the dimensionality $d$ and the number of samples $n$ tend to infinity. This high-dimensional setting is widely studied as it can often – as in our experiments – yield precise predictions for the risk of the estimator when both the input dimension and the data set size are large [9, 52]. It is also the predominant setting considered in previous theoretical works that discuss overparameterized linear models [2, 14, 15, 23, 25, 24, 50].

## 2.2 Standard and robust risk

We now introduce the standard and robust evaluation metrics for regression and classification. Given a pointwise test loss $\ell_{\text{test}} \colon \mathbb{R} \times \mathbb{R} \to \mathbb{R}$, we define the standard (population) risk of an estimator $\hat{\theta}$ as

$$\mathbf{R}(\hat{\theta}) := \mathbb{E}_{X \sim \mathbb{P}} \ell_{\text{test}} \left( \langle \hat{\theta}, X \rangle, \langle \theta^\star, X \rangle \right), \tag{1}$$

where the expectation is taken over the marginal feature distribution $\mathbb{P}$. Note that for any data-dependent estimator $\hat{\theta}$, this risk is fixed if conditioned on the training data. Our asymptotic bounds hold almost surely over draws of the training set. As standard in the literature, we choose the square loss $\ell_{\text{test}}(u, v) = (u - v)^2$ for regression and the 0-1 loss $\ell_{\text{test}}(u, v) = \mathbb{1}_{\text{sgn}(u) \neq \text{sgn}(v)}$ for classification.

The broad application of ML models in real-world decision-making processes increases requirements on their generalization abilities beyond i.i.d. test sets. For example, in the image domain, classifiers should be *robust* and output the same prediction for perturbations of an image that do not change the ground truth label (e.g., imperceptible $\ell_p$-perturbations [19]). In this case, we say the perturbations are *consistent* and the estimator that achieves zero robust population risk also has zero standard population risk. For linear models in particular, one way to enforce consistency is to restrict perturbations to the space orthogonal to the ground truth, as proposed in [41].

Motivated by the imperceptibility assumption and $\ell_p$-adversarial attacks widely studied in the image domain, we consider the adversarially robust risk of a parameter $\theta$ with respect to consistent $\ell_p$-perturbations

$$\mathbf{R}_\epsilon(\hat{\theta}) := \mathbb{E}_{X \sim \mathbb{P}} \max_{\delta \in \mathcal{U}_p(\epsilon)} \ell_{\text{test}}(\langle \hat{\theta}, X + \delta \rangle, \langle \theta^\star, X \rangle), \tag{2}$$

with the perturbation set $\mathcal{U}_p(\epsilon) := \{\delta \in \mathbb{R}^d : \|\delta\|_p \leq \epsilon \text{ and } \langle \theta^\star, \delta \rangle = 0\}$.

In many scientific applications, security against adversarial attacks may not be the dominating concern; one may instead require estimators to be robust against small distribution shifts. Earlier work [48] has pointed out that distribution shift robustness and adversarial robustness are equivalent for losses that are convex in the parameter $\theta$. Similarly, in our setting, adversarial robustness against consistent $\ell_p$-perturbations implies distributional robustness against $\ell_p$-bounded mean shifts in the covariate distribution $\mathbb{P}$ (see Appendix A.3).

## 3 Min-$\ell_2$-norm interpolation in robust linear regression

In the context of regression, we illustrate overfitting of the robust risk in Equation (2) with the set of consistent $\ell_2$-perturbations $\mathcal{U}_2(\epsilon)$. More precisely, we show that preventing min-$\ell_2$-norm interpolation on noiseless samples via ridge regularization improves the robust risk. We refer the reader to Appendix C.1 for an intuitive explanation. Lastly, we note that due to the rotational invariance of the problem, our results hold for sparse and dense ground truths $\theta^\star$ alike.

### 3.1 Interpolating and regularized estimator

We study linear ridge regression estimates defined as

$$\hat{\theta}_\lambda = \arg\min_\theta \frac{1}{n}\sum_{i=0}^{n}(y_i - \langle\theta, x_i\rangle)^2 + \lambda\|\theta\|_2^2. \tag{3}$$

The min-$\ell_2$-norm interpolator is the limit of the linear ridge regression estimate with $\lambda \to 0$ and is given by

$$\hat{\theta}_0 = \arg\min_\theta \|\theta\|_2 \ \text{ such that } \ \langle\theta, x_i\rangle = y_i \ \text{ for all } i. \tag{4}$$

Note that the min-$\ell_2$-norm interpolator is also the estimator that gradient descent on the unregularized loss converges to, while ridge regression with $\lambda > 0$ corresponds to early-stopped estimators [1, 2]. Therefore, by proving that a ridge regularized estimator with $\lambda > 0$ significantly outperforms the min-$\ell_2$-norm interpolator with $\lambda \to 0$, we also show that early stopping benefits robust generalization.

Whenever the goal is to achieve a low robust risk, a popular alternative to using the standard linear regression estimate in Equations (3),(4) is to consider adversarially trained estimators [19, 25]. However, $\ell_2$-adversarial training in its usual form (i.e., with inconsistent perturbations) prevents regression estimators from interpolating, and hence, has a similar effect to $\ell_2$-regularization as we discuss in more detail in Appendix C.2. On the other hand, training with consistent perturbations as defined in the robust risk is equivalent to full knowledge of the direction of $\theta^\star$ and hence simply recovers the ground truth in the noiseless case. Since our goal is to reveal the shortcomings of interpolators compared to regularized estimators, we only analyze ridge estimators trained without perturbations.

### 3.2 Robust overfitting in noiseless linear regression

The following theorem provides a precise asymptotic expression of the robust risk under consistent $\ell_2$-perturbations for the ridge regression estimate in Equation (3). The proof extends techniques from previous works [23, 15] based on results from random matrix theory [3, 28] and can be found in Appendix E. Without loss of generality, we can assume that $\|\theta^\star\|_2 = 1$.

**Theorem 3.1.** *Assuming the marginal input distribution $\mathbb{P} = \mathcal{N}(0, I_d)$, the robust risk (2) of the estimator $\hat{\theta}_\lambda$ for $\lambda > 0$ (defined in (3)) with respect to consistent $\ell_2$-perturbations $\mathcal{U}_2(\epsilon)$ asymptotically converges to*

$$\boldsymbol{R}_\epsilon(\hat{\theta}_\lambda) \xrightarrow{a.s.} \mathcal{R}_\lambda + \epsilon^2\mathcal{P}_\lambda + \sqrt{\frac{8\epsilon^2}{\pi}\mathcal{P}_\lambda\mathcal{R}_\lambda} =: \mathcal{R}_{\epsilon,\lambda} \tag{5}$$

*as $d, n \to \infty$ with $d/n \to \gamma$, where $\mathcal{P}_\lambda = \mathcal{R}_\lambda - \lambda^2(m(-\lambda))^2$ and $\mathcal{R}_\lambda = \lambda^2 m'(-\lambda) + \sigma^2\gamma(m(-\lambda) - \lambda m'(-\lambda))$ is the asymptotic standard risk, i.e., $\boldsymbol{R}(\hat{\theta}_\lambda) \xrightarrow{a.s.} \mathcal{R}_\lambda$. The function $m(z)$ is the Stieltjes transform of the Marchenko-Pastur law and is given by $m(z) = \frac{1-\gamma-z-\sqrt{(1-\gamma-z)^2-4\gamma z}}{2\gamma z}$. Further, the limit $\lim_{\lambda\to 0}\mathcal{R}_{\epsilon,\lambda}$ exists for all $\epsilon \geq 0$ and corresponds to the asymptotic standard ($\epsilon = 0$) and robust risks ($\epsilon > 0$) of the min-$\ell_2$-norm interpolator $\hat{\theta}_0$ (4).*

We plot the precise asymptotic risks of the ridge estimate with optimal regularization parameter $\lambda_{\text{opt}}$[2] and of the min-$\ell_2$-norm interpolator for $\lambda \to 0$ in Figure 2. For the robust risk, we use $\epsilon = 0.4$. We first observe in Figure 2a that ridge regularization reduces the robust risk even for $d/n \gg 1$ well beyond the interpolation threshold – the regime where previous works show that the variance is negligible, and hence, regularization does not improve generalization.

Moreover, Figure 2b illustrates that the beneficial effect of ridge regularization persists even for noiseless data. This supports our statement that regularization not only helps to reduce variance, but also reduces the part of the robust risk that is unaffected by noise in the overparameterized regime. Furthermore, we show that experiments run with finite values of $d$ and $n$ (depicted by the markers in Figure 2) closely match the predictions obtained from Theorem 3.1 for $d, n \to \infty$ and $d/n \to \gamma$. This

---

[2]Here we choose $\lambda$ using the population risk oracle. In practice, one would resort to standard tools such as cross-validation techniques that also enjoy theoretical guarantees (see e.g. [38]).

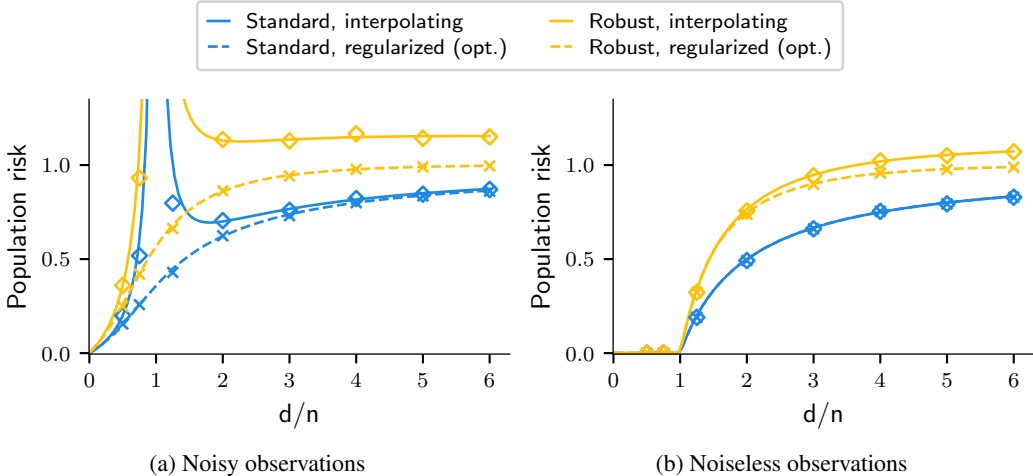

(a) Noisy observations           (b) Noiseless observations

Figure 2: Asymptotic theoretical predictions for $d, n \to \infty$ (curves) and experimental results with finite $d$ and $n = 10^3$ (markers) for the robust ($\epsilon = 0.4$) and standard risk of the min-$\ell_2$-norm interpolator (solid, *interpolating*) and the ridge regression estimate with optimal $\lambda$ (dashed, *regularized*) for (a) noisy data with $\sigma^2 = 0.2$ and (b) noiseless data. We observe that the gap between the robust risk of the interpolating and optimally regularized estimators persists even for noiseless observations.

indicates that the high-dimensional asymptotic regime does indeed correctly predict and characterize the high-dimensional non-asymptotic regime. Finally, even though Theorem 3.1 assumes isotropic Gaussian covariates, we can extend it to more general covariance matrices following the same argument as in [23], based on results from random matrix theory [28].

## 4    Max-$\ell_2$-margin interpolation in robust linear classification

Unlike linear regression, adversarially trained binary logistic regression classifiers may still interpolate the training data, resulting in *robust* max-$\ell_2$-margin interpolators as $\lambda \to 0$. Hence, in this section we train and evaluate classifiers with $\ell_\infty$-perturbation sets $\mathcal{U}_\infty(\epsilon)$, a standard choice in the experimental and theoretical classification literature [19, 24, 42, 47], but also discuss $\ell_2$-perturbations in Appendix D.3 for completeness. Our theoretical results show that the robust max-$\ell_2$-margin interpolator with $\lambda \to 0$ has a worse robust risk than a regularized predictor with $\lambda > 0$.

### 4.1    Interpolating and regularized estimator

As discussed in Section 3, a common method to obtain robust estimators is to use adversarial training. However, for linear regression, adversarial training either renders interpolating estimators infeasible, or requires oracle knowledge of the ground truth. In contrast, for linear classification, interpolation is easier to achieve – it only requires the sign of $\langle x_i, \theta \rangle$ to be the same as the label $y_i$ for all $i$. In particular, when the data is sufficiently high dimensional, it is possible to find an interpolator of the adversarially perturbed training set.

We study the robust ridge-regularized logistic regression estimator with penalty weight $\lambda > 0$,

$$\hat{\theta}_\lambda := \arg\min_\theta \frac{1}{n} \sum_{i=1}^n \max_{\delta \in \mathcal{U}_\infty(\epsilon)} \log(1 + e^{-\langle \theta, x_i + \delta \rangle y_i}) + \lambda \|\theta\|_2^2. \tag{6}$$

In the limit $\lambda \to 0$ the results in [44] imply that the robust ridge-regularized logistic regression estimator from Equation (6) directionally aligns with the robust max-$\ell_2$-margin interpolator:[3]

$$\hat{\theta}_0 := \arg\min_\theta \|\theta\|_2 \text{ such that } \min_{\delta \in \mathcal{U}_\infty(\epsilon)} y_i \langle \theta, x_i + \delta \rangle \geq 1 \text{ for all } i. \tag{7}$$

---

[3]While [44] only proves the result for $\epsilon = 0$, it is straightforward to extend it to the general case where $\epsilon \geq 0$.

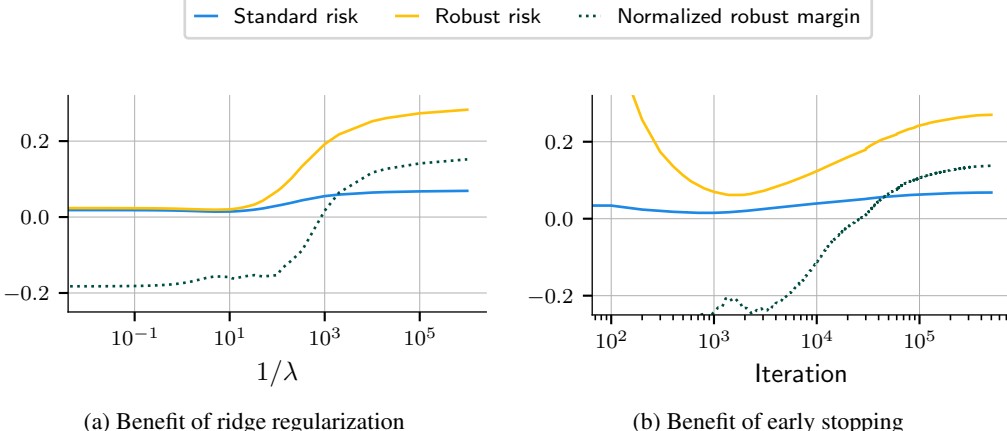

(a) Benefit of ridge regularization                    (b) Benefit of early stopping

Figure 3: Normalized robust margins and risks of empirical simulations using $\epsilon = 0.1$ and $d/n = 8$, with respect to (a) increasing $1/\lambda$ and (b) gradient descent iterations when minimizing Equation (6) using $\lambda = 0$. Both ridge regularization and early stopping yield superior robust and standard risks. Each experiment uses $n = 10^3$ and inconsistent $\ell_\infty$-perturbations for training. See Appendix B for more details.

We say that the data is *robustly separable* if the robust max-$\ell_2$-margin interpolator exists.

The robust max-$\ell_2$-margin solution is an interpolating estimator of particular importance since it directionally aligns with the estimator found by gradient descent [31]. Since the robust accuracy (i.e., the robust risk defined using the 0-1 loss) is independent of the norm of the estimator, we simply refer to the robust max-$\ell_2$-margin solution as the normalized vector $\frac{\hat{\theta}_0}{\|\hat{\theta}_0\|_2}$.

In this paper, we study two choices for the set of training perturbations $\mathcal{U}_\infty(\epsilon)$:

$$\text{inconsistent perturbations} \qquad \mathcal{U}_\infty(\epsilon) = \{\delta \in \mathbb{R}^d : \|\delta\|_\infty \le \epsilon\} \qquad (8)$$

$$\text{consistent perturbations} \qquad \mathcal{U}_\infty(\epsilon) = \{\delta \in \mathbb{R}^d : \|\delta\|_\infty \le \epsilon, \langle \delta, \theta^\star \rangle = 0\} \qquad (9)$$

Adversarial training with respect to inconsistent perturbations (8) is a popular choice in the literature to improve the robust risk (e.g. [19, 24]). However, perturbed samples may cross the true decision boundary, and hence, inconsistent perturbations effectively introduce noise during the training procedure. In particular, in the data model with noiseless observations that we introduce in Section 4.2, the ground truth function misclassifies approximately $8\%$ of the labels when perturbing the training data with inconsistent perturbations of size $\epsilon = 0.1$.

As mentioned in the introduction, in this paper we are interested in verifying whether regularization can be beneficial in high dimensions even in the absence of noise. Therefore, in the sequel we study the impact of both inconsistent (8) and consistent (9) perturbations on robust overfitting.

## 4.2 Robust overfitting in noiseless linear classification

We now show empirically that regularization helps to improve the robust and standard risks when training with noiseless data and derive precise asymptotic predictions for both risks. Throughout this subsection we assume deterministic, and hence, noiseless training labels, i.e., $y_i = \text{sgn}\langle x_i, \theta^\star \rangle$. Furthermore, as we discuss in Section 4.3, the inductive bias of the $\ell_\infty$-robust logistic loss encourages sparse solutions. Since we are primarily interested in learning ground truth functions that match the implicit bias of the estimator, we assume the sparse ground truth $\theta^\star = (1, 0, \ldots, 0)^T$.

We first show robust overfitting experimentally on noiseless data when training with inconsistent perturbations and subsequently demonstrate that overfitting persists even if the training procedure is completely noiseless (i.e., using consistent training perturbations). Finally, we provide theoretical evidence for our observations in the high-dimensional asymptotic limit.

**Training with inconsistent adversarial perturbations** Figure 3a illustrates the *robust margin* $\min_i \min_{\delta \in \mathcal{U}_\infty(\epsilon)} \frac{1}{\|\theta\|_2} y_i \langle \theta, x_i + \delta \rangle$ as well as the standard and robust risks of the estimator $\hat{\theta}_\lambda$ trained using inconsistent adversarial perturbations on a synthetic data set with fixed overparameterization ratio $d/n = 8$. We observe that decreasing the ridge coefficient well beyond the point where the minimizer of the robust logistic loss (6) reaches $100\%$ robust training accuracy (i.e., the robust margin becomes positive) substantially hurts generalization.

In addition to varying the ridge coefficient $\lambda$, we notice that the same trends as for $\lambda \to 0$ also occur for the gradient descent optimization path as the number of iterations $t$ goes to infinity. Figure 3b indicates that, similarly to ridge regularization, early stopping also avoids the robust max-$\ell_2$-margin solution that is obtained for $t \to \infty$ and yields an estimator with significantly lower standard and robust risks.

**Training with consistent adversarial perturbations** As discussed in Section 4.1, even for noiseless training data, inconsistent perturbations can induce noise during the training procedure. Hence, one could hypothesize that the noise induced by the inconsistent perturbations causes the overfitting observed in Figure 3. To contradict this hypothesis, we also study adversarial training with consistent perturbations. By definition, consistent perturbations do not cross the true decision boundary and hence leave the training data entirely noiseless.

Figure 4a shows that the adversarially trained estimators (6),(7) with consistent and inconsistent perturbations yield comparable robust risks. Moreover, robust overfitting occurs in both situations, as the risk is higher for the interpolating estimator $\lambda \to 0$ compared to an optimal $\lambda > 0$. Hence, our observations demonstrate that robust overfitting persists even if training with consistent perturbations in an entirely noiseless setting. This observation is counter intuitive since, according to classical wisdom, we would expect ridge regularization to only benefit in noisy settings where the estimator suffers from a high variance.

We now prove this phenomenon using the next theorem. In particular, similar to Theorem 3.1 for linear regression, we show that robust overfitting occurs in the high-dimensional asymptotic regime where $d/n \to \gamma$ as $d, n \to \infty$. We state an informal version of the theorem in the main text and refer to Appendix F for the precise statement. The proof is inspired by the works [24, 46] and uses the Convex Gaussian Minimax Theorem (CGMT) [20, 51].

**Theorem 4.1** (Informal). *Assume that $\epsilon = \epsilon_0/\sqrt{d}$ for some constant $\epsilon_0$ and $\theta^\star = (1, 0, \cdots, 0)^T$. Then, the robust and standard risks of the regularized estimator $\hat{\theta}_\lambda$ (6) ($\lambda > 0$) and of the robust max-$\ell_2$-margin interpolator (7) ($\lambda \to 0$) with inconsistent (8) or consistent (9) adversarial $\ell_\infty$-perturbations converge in probability as $d, n \to \infty$, $d/n \to \gamma$ to:*

$$\boldsymbol{R}(\hat{\theta}_\lambda) \to \frac{1}{\pi} \arccos\left(\frac{\nu_\parallel^\star}{\nu^\star}\right) \;\; \text{and} \;\; \boldsymbol{R}_\epsilon(\hat{\theta}_\lambda) \to \boldsymbol{R}(\hat{\theta}_\lambda) + \frac{1}{2}\mathrm{erf}\left(\frac{\epsilon_0 \delta^\star}{\sqrt{2}\nu^\star}\right) + I\left(\frac{\epsilon_0 \delta^\star}{\nu^\star}, \frac{\nu_\parallel^\star}{\nu^\star}\right)$$

*We denote by* $\mathrm{erf}(.)$ *the error function,*

$$I(t, u) := \int_0^t \frac{1}{\sqrt{2\pi}} \exp\left(-\frac{x^2}{2}\right) \mathrm{erf}\left(\frac{xu}{\sqrt{2(1 - u^2)}}\right) dx,$$

*and use the notation* $\nu^\star = \sqrt{(\nu_\perp^\star)^2 + (\nu_\parallel^\star)^2}$, *where* $\nu_\perp^\star, \nu_\parallel^\star, \delta^\star$ *are the unique solution of a scalar optimization problem specified in Appendix F that depends on* $\theta^\star, \gamma, \epsilon_0$ *and* $\lambda$.

Since the theoretical expressions are hard to interpret, we visualize the asymptotic values of the standard and robust risks from Theorem 4.1 in Figure 4b by solving the scalar optimization problem specified in Appendix F. We observe that Theorem 4.1 indeed predicts the benefits of regularization for robust logistic regression and that simulations using finite values of $d$ and $n$ follow the asymptotic trend. We describe the full empirical setup in Appendix B.

### 4.3 Intuitive explanation and discussion

Even though we explicitly derive the precise asymptotic expressions of the standard and robust risks in Theorem 4.1 that predict the benefits of regularization for generalization, it is difficult to extract intuitive explanations for this phenomenon directly from the proof. We conjecture that a non-zero

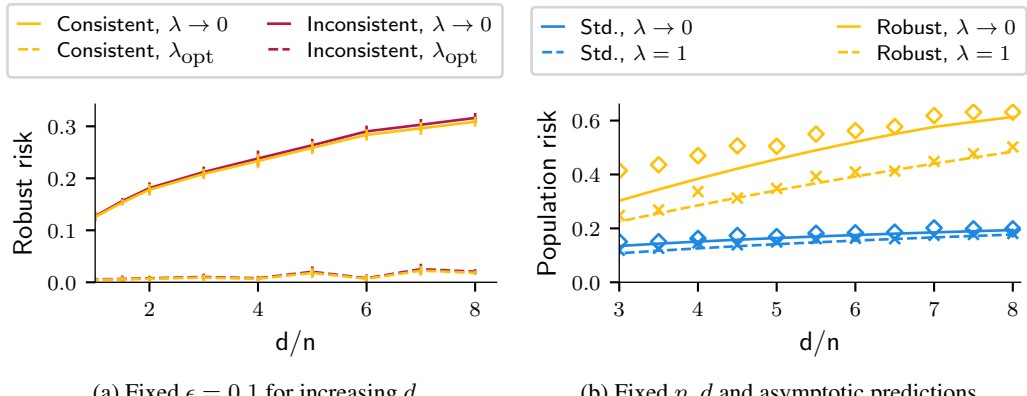

(a) Fixed $\epsilon = 0.1$ for increasing $d$

(b) Fixed $n, d$ and asymptotic predictions

Figure 4: (a) Comparison of consistent and inconsistent $\ell_\infty$-perturbations for adversarial logistic regression with respect to the degree of overparameterization $d/n$, using $\epsilon = 0.1$ for both training and evaluation. Note that both estimators behave very similarly, implying that the effect of inconsistency is negligible for small $\epsilon$. (b) Robust and standard risks of the robust max-$\ell_2$-margin interpolator ($\lambda \to 0$) and robust ridge estimate ($\lambda = 1$) with consistent perturbations (9) using $\epsilon = 0.05$ as a function of the overparameterization ratio $d/n$ for simulations (markers) and asymptotic theoretical predictions from Theorem 4.1 (lines). We note that, for small values of $\gamma$, solving the optimization problem that gives the theoretical predictions becomes numerically unstable. All simulations use $n = 10^3$ samples from our data model; see Appendix B for further experimental details.

ridge penalty induces a more sparse $\hat{\theta}_\lambda$ (i.e., with a smaller $\ell_1/\ell_2$-norm ratio) than the robust max-$\ell_2$-margin solution $\hat{\theta}_0$ and use simulations to support our claim. Since the $\ell_\infty$-adversarially robust risk penalizes dense solutions with large ratio of the $\ell_1/\ell_2$-norms (see Lemma A.2 in Appendix A.2), we expect more sparse estimators to have a lower robust risk. Indeed, Figure 5a shows that the $\ell_1/\ell_2$-norm ratio strongly correlates with the robust risk of the estimator.

We begin by noting that, due to Lagrangian duality, minimizing the ridge-penalized loss (6) corresponds to minimizing the unregularized loss constrained to the set of estimators $\theta$ with a bounded $\ell_2$-norm. This norm decreases as the ridge coefficient $\lambda$ increases. In what follows, we provide intuition on the effect of the ridge penalty $\lambda$ on the sparsity of the estimator $\hat{\theta}_\lambda$.

**Large $\lambda$ inducing a small $\ell_2$-norm** We first analyze the regularized estimator $\hat{\theta}_\lambda$ for large $\lambda$ that constrains solutions to have small $\ell_2$-norm. We can therefore use Taylor's theorem and the closed-form expression of adversarial perturbations (see Lemma A.2 in Appendix A.2) to approximate the unregularized robust loss from Equation (6) as follows:

$$\frac{1}{n}\sum_{i=1}^n \log(1 + e^{-y_i\langle\theta,x_i\rangle + \epsilon\|\Pi_\perp\theta\|_1}) \approx \frac{1}{n}\sum_{i=1}^n -y_i\langle x_i,\theta\rangle + \epsilon\|\Pi_\perp\theta\|_1. \tag{10}$$

As a consequence, the minimizer $\hat{\theta}_\lambda$ should result in a large *robust average margin* solution, that is, a solution with large $\frac{1}{n}\sum_{i=1}^n \frac{1}{\|\theta\|_2}(y_i\langle x_i,\theta\rangle - \epsilon\|\Pi_\perp\theta\|_1)$. Indeed, we observe this using simulations for finite $d, n$ in Figures 5b and 5c. In particular, the objective in Equation (10) leads to a trade-off between the sparsity of the estimator (via its convex surrogate, the $\ell_1$-norm) and an *average* of the sample-wise margins $y_i\langle x_i,\theta\rangle$. We note that such estimators have been well studied in the literature and are known to achieve good performance in recovering sparse ground truths [5, 17, 39].

**Small $\lambda$ inducing a large $\ell_2$-norm** In contrast, a small ridge coefficient $\lambda$ leads to estimators $\hat{\theta}_\lambda$ with large $\ell_2$-norms. In this case, the estimator approaches a large *robust (minimum) margin* solution, i.e., a solution with large $\min_i \frac{1}{\|\theta\|_2}(y_i\langle x_i,\theta\rangle - \epsilon\|\Pi_\perp\theta\|_1)$ which is maximized by the robust max-$\ell_2$-margin interpolator (7). As a consequence, this leads to a trade-off between estimator sparsity and the robust minimum margin $\min_i \frac{1}{\|\theta\|_2}y_i\langle x_i,\theta\rangle$. Due to the high dimensionality of the input data, the training samples $x_i$ are approximately orthogonal. Thus, to achieve a non-vanishing robust margin, estimators are forced to trade-off sparsity with *all* sample-wise margins instead of just

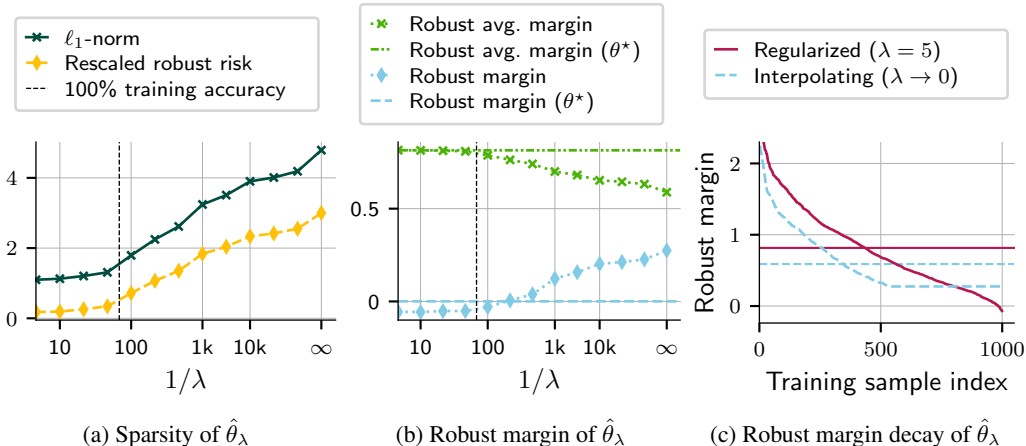

(a) Sparsity of $\hat{\theta}_\lambda$      (b) Robust margin of $\hat{\theta}_\lambda$      (c) Robust margin decay of $\hat{\theta}_\lambda$

Figure 5: (a) The $\ell_1$-norm and the rescaled (by a factor of 10) robust risk of the estimator with respect to $1/\lambda$. (b) The robust average margin contrasted to the robust margin as a function of $1/\lambda$. The horizontal lines denote the corresponding values for $\theta^\star$. (c) The ordered sample-wise robust margins $y_i\langle x_i, \theta\rangle - \epsilon\|\Pi_\perp\theta\|_1$ when interpolating and regularizing. For larger $\lambda$, the robust (*minimum*) margin decreases while the robust average margin (horizontal lines) increases. We normalize the estimators, i.e., $\|\hat{\theta}_\lambda\|_2 = 1$, for all curves presented in the plots; see Appendix B for further experimental details.

the average. We reveal this trade-off in Figures 5a and 5b where the increase in $\ell_1/\ell_2$-norm ratios corresponds to a decrease in the robust average margin and an increase in the robust margin.

Finally, we observe that the *sparse* ground truth is characterized by a large robust average margin (horizontal dotted line in Figure 5b) and a small (minimum) robust margin (horizontal dashed line). Therefore, we expect that the solution that is sparser and which satisfies the same properties for the robust margin as the ground truth $\theta^\star$, will achieve lower robust and standard risks. Indeed, our findings indicate that the regularized estimator $\hat{\theta}_\lambda$ for large $\lambda$ aligns better with $\theta^\star$, compared to the solution obtained for a small $\lambda$, and hence justify the better performance of ridge-regularized predictors.

### 4.4 Benefits of an unorthodox way to avoid the robust max-$\ell_2$-margin interpolator

In the previous subsections we focused on robustly separable data and studied the generalization performance of regularized estimators that do not maximize the robust margin. Another way to avoid the robust max-$\ell_2$-margin solution is to introduce enough label noise in the training data. We now show that, unexpectedly, this unorthodox way to avoid the robust max-margin solution can also yield an estimator with better robust generalization than the robust max-$\ell_2$-margin solution of the corresponding noiseless problem.

Specifically, in our experiments we introduce noise by flipping the labels of a fixed fraction of the training data. Figure 6 shows the robust and standard risks together with the training loss of the estimator $\hat{\theta}_\lambda$ from Equation (6) trained with consistent perturbations for $\lambda \to 0$ with varying fractions of flipped labels. For low noise levels, the data is robustly separable and the training loss vanishes at convergence, yielding the robust max-$\ell_2$-margin solution in Equation (7). For high enough noise levels, the constraints in Equation (7) become infeasible and the training loss of the resulting estimator starts to increase. As discussed in Subsection 4.3, this estimator has a better implicit bias than the robust max-$\ell_2$-margin interpolator and hence achieves a lower robust risk.

Even though it is well known that introducing covariate noise can induce implicit regularization [8], our observations show that in contrast to common intuition, the robust risk also decreases when introducing wrong labels in the training loss. In parallel to our work, the paper [30] shows that training with corrupted labels can be beneficial for the standard risk.

However, we emphasize here that we do not advocate in favor of artificial label noise as a means to obtain more robust classifiers. In particular, even if the data is not robustly separable, the estimator with optimal ridge parameter $\lambda_{\text{opt}}$ in Figure 6 still always outperforms the unregularized solution.

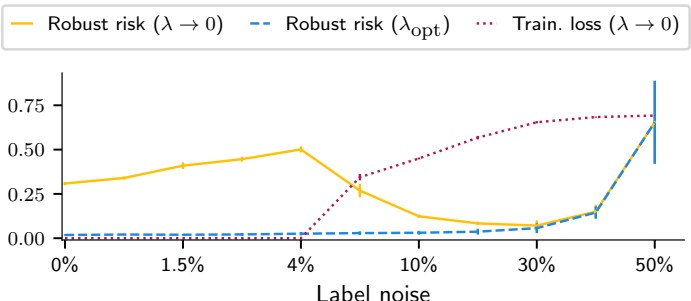

Figure 6: Training loss and robust risks with respect to increasing training label noise for $\epsilon = 0.1$, $d = 8 \times 10^3$ and $n = 10^3$. We observe for unregularized estimators ($\lambda \to 0$) that, counterintuitively, moderate amounts of label noise decrease the robust risk by avoiding the robust max-$\ell_2$-margin solution. While this might spuriously imply that injecting label noise increases robustness, estimators with optimal ridge parameter $\lambda_{\text{opt}}$ still outperform their unregularized counterparts in terms of robust risk. Since the setting is noisy, we average the risks over five independent dataset draws and indicate standard deviations via error bars.

Finally, we remark that a similar effect can also be observed when training with inconsistent perturbations with large perturbation norm $\epsilon$. We refer to Appendix D.1 for further discussion.

## 5 Related work

**Understanding robust overfitting**  The current literature attempting to explain robust overfitting [42] primarily focuses on the effect of noise and on the smoothness of decision boundaries learned by neural networks trained to convergence [16, 47, 53]. A slightly different line of work [45] shows that overparameterized models require regularization in order to achieve good classification accuracy on all subpopulations. However, a theoretical understanding of the role of regularization is still missing.

**Theory for adversarial robustness of linear models**  Recent works [25, 24] provide a precise description of the robust risks for logistic and linear regression when trained with adversarial attacks based on the Convex Gaussian Minimax Theorem [20, 51]. The analysis focuses on inconsistent attacks for both training and evaluation. For linear regression, the authors observe that adversarial training with $\ell_2$-perturbations mitigates the peak in the double descent curve around $d = n$ and hence acts similarly to ridge regularization as explicitly studied in [23]. Several other works focus on the role of gradient descent in robust logistic regression. In particular, [56] proves that early stopping yields robust adversarially-trained linear classifiers even in the presence of noise. Furthermore, the results of [31] show that gradient descent on robustly separable data converges to the robust max-$\ell_2$-margin estimator (7).

## 6 Conclusion and future work

In this work, we show that overparameterized linear models can overfit with respect to the robust risk even when there is no noise in the training data. Our results challenge the modern narrative that interpolating overparameterized models yield good performance without explicit regularization and motivate the use of ridge regularization and early stopping for improved robust generalization. Perhaps surprisingly, we further observe that ridge regularization enhances the bias of logistic regression trained with adversarial $\ell_\infty$-attacks towards sparser solutions, indicating that the impact of explicit regularization may go well beyond variance reduction.

**Future work**  Our simulations indicate that early stopping yields similar benefits as ridge regularization in noiseless settings. However, we leave a formal proof for future work. Furthermore, the double descent phenomenon has been proven for the standard risk on a broad variety of data distributions and even for non-linear models such as random feature regression. It is still unclear how our results translate to these settings. In particular, our theoretical analysis heavily makes use of the closed-form solution of the adversarial attacks, and hence, cannot be applied to non-linear models in a straightforward way.

## Acknowledgments

K. D. is supported by the ETH AI Center and the ETH Foundations of Data Science. R. H. is supported by the IAS at TUM, the DFG (German Research Foundation), and by the NSF under award IIS-1816986.

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
