# A Closed form expressions for the robust risks

In Section A.1 and A.2 we derive closed-form expressions of the standard and robust risks from Equations (1),(2) for the settings studied in Section 3,4. We use those expressions repeatedly in our proofs. Furthermore, Section A.3 discusses that the robust risk (2) upper-bounds the worst case risk under distributional mean shifts.

## A.1 Closed-form of robust risk for regression

The following lemma provides a closed-form expression of the robust risk for the linear regression setting studied in Section 3. A similar result for inconsistent attacks has already been shown before (Lemma 3.1. in [25]); we only include the proof for completeness.

**Lemma A.1.** *Assume that $\mathbb{P}_X$ is the isotropic Gaussian distribution. Then, for the square loss, the robust risk* (2) *with respect to $\ell_2$-perturbations is given by*

$$\boldsymbol{R}_\epsilon(\theta) = \|\theta^\star - \theta\|_2^2 + 2\epsilon\sqrt{2/\pi}\|\Pi_\perp\theta\|_2\|\theta^\star - \theta\|_2 + \epsilon^2\|\Pi_\perp\theta\|_2^2. \tag{11}$$

*Proof.* Define $\tilde{y}_i = y_i - \langle x_i, \theta\rangle$, and note that using similar arguments as in Section 6.2. [25]

$$\max_{\delta_i \in \mathcal{U}_2(\epsilon)} (\tilde{y}_i - \langle \delta_i, \theta\rangle)^2 = (\max_{\delta_i \in \mathcal{U}_2(\epsilon)} |\tilde{y}_i - \langle \delta_i, \theta\rangle|)^2$$
$$= (|\tilde{y}_i| + \max_{\|\delta_i\|_2 \leq \epsilon, \delta_i \perp \theta^\star} |\langle \delta_i, \theta\rangle|)^2$$
$$= (|\tilde{y}_i| + \epsilon\|\Pi_\perp\theta\|_2)^2.$$

With this characterization, we can derive a convenient expression for the robust risk:

$$\mathbf{R}_\epsilon(\theta) = \mathbb{E}_X(|\langle X, \theta^\star - \theta\rangle| + \epsilon\|\Pi_\perp\theta\|_q)^2$$
$$= \mathbb{E}_X(\langle X, \theta^\star - \theta\rangle)^2 + 2\epsilon\mathbb{E}_X|\langle X, \theta^\star - \theta\rangle|\|\Pi_\perp\theta\|_2 + \epsilon^2\|\Pi_\perp\theta\|_2^2. \tag{12}$$

Since we assume isotropic Gaussian features, that is $\mathbb{P}_X = \mathcal{N}(0, I)$, we can further simplify

$$\mathbf{R}_\epsilon(\theta) = \|\theta - \theta^*\|_2^2 + 2\epsilon\sqrt{2/\pi}\|\Pi_\perp\|_2\|\theta - \theta^*\|_2 + \epsilon^2\|\Pi_\perp\|_2^2$$

which concludes the proof. $\qquad\square$

## A.2 Closed-form of robust risk for classification

Similarly to linear regression, we can express the robust and standard risk for the linear classification model in Section 4 as stated in the following lemma.

**Lemma A.2.** *Assume that $\mathbb{P}_X$ is the isotropic Gaussian distribution and $\theta^\star = (1, 0, \cdots, 0)^\top$. Then,*

1. *For any non-decreasing loss $\ell : \mathbb{R} \to \mathbb{R}$ we have*

$$\max_{\delta_i \in \mathcal{U}_\infty(\epsilon)} \ell(y_i \langle x_i + \delta_i, \theta\rangle) = \ell(y_i \langle x_i, \theta\rangle - \epsilon\|\Pi_\perp\theta\|_1). \tag{13}$$

2. *For the 0-1 loss the robust risk* (2) *with respect to $\ell_\infty$-perturbations is given by*

$$\boldsymbol{R}_\epsilon(\theta) = \frac{1}{\pi}\arccos\left(\frac{\langle\theta^\star, \theta\rangle}{\|\theta\|_2}\right) + \frac{1}{2}\mathrm{erf}\left(\frac{\epsilon\|\Pi_\perp\theta\|_1}{\sqrt{2}\|\theta\|_2}\right) + I\left(\frac{\epsilon\|\Pi_\perp\theta\|_1}{\|\theta\|_2}, \frac{\langle\theta^\star, \theta\rangle}{\|\theta\|_2}\right), \tag{14}$$

*with*

$$I(t, u) := \int_0^t \frac{1}{\sqrt{2\pi}}\exp\left(-\frac{x^2}{2}\right)\Phi\left(\frac{xu}{\sqrt{2}\sqrt{1-u^2}}\right)dx. \tag{15}$$

*Proof.* We first prove Equation (13). Because $\ell$ is non-increasing, we have

$$\max_{\delta_i \in \mathcal{U}_\infty(\epsilon)} \ell(y_i \langle x_i + \delta_i, \theta\rangle) = \ell(\min_{\delta_i \in \mathcal{U}_\infty(\epsilon)} y_i \langle x_i + \delta_i, \theta\rangle)$$
$$= \ell(y_i \langle x_i, \theta\rangle + \min_{\|\delta_i\|_\infty \leq \epsilon, \delta_i \perp \theta^\star} \langle \delta_i, \theta\rangle)$$
$$= \ell(y_i \langle x_i, \theta\rangle - \epsilon\|\Pi_\perp\theta\|_1),$$

which establishes Equation (13). Note that for the last equation we used that, while minimization over $\delta$ has no closed-form solution in general, for our choice of $\theta^\star = (1, 0, \cdots, 0)$, we get $\min_{\|\delta_i\|_\infty \leq \epsilon, \delta_i \perp \theta^\star} \langle \delta_i, \theta \rangle = -\epsilon \|\Pi_\perp \theta\|_1$.

We now prove the second part of the statement. Let $\mathbb{1}\{E\}$ be the indicator function, which is 1 if the event $E$ occurs, and 0 otherwise. Since $\ell(\cdot) = \mathbb{1}_{\cdot \leq 0}$ is non-increasing we can use (13) and write

$$\mathbf{R}_\epsilon(\theta) = \mathbb{E}_X \max_{\delta \in \mathcal{U}_\infty(\epsilon)} \mathbb{1}\{\operatorname{sgn}(\langle X, \theta^\star \rangle)\langle X + \delta, \theta \rangle \leq 0\}$$

$$= \mathbb{E}_X \mathbb{1}\{\operatorname{sgn}(\langle X, \theta^\star \rangle)\langle X, \theta \rangle - \epsilon \|\Pi_\perp \theta\|_1 \leq 0\}.$$

Let $\widehat{\Pi}_\| := \frac{1}{\|\theta\|_2^2}\theta\theta^\top$ be the projection onto the subspace spanned by $\theta$ and $\widehat{\Pi}_\perp := I_d - \widehat{\Pi}_\|$ the projection onto the orthogonal complement. Since $X$ is a vector with i.i.d. standard normal distributed entries, we equivalently have

$$\mathbf{R}_\epsilon(\theta) = \mathbb{E}_{Z_1, Z_2} \mathbb{1}\{Z_1 \operatorname{sgn}\left(Z_1 \|\widehat{\Pi}_\| \theta^\star\|_2 + Z_2 \|\widehat{\Pi}_\perp \theta^\star\|_2\right) - \epsilon \frac{\|\Pi_\perp \theta\|_1}{\|\theta\|_2} \leq 0\}, \qquad (16)$$

with $Z_1, Z_2$ two independent standard normal random variables. For brevity of notation, define $\nu = \epsilon \frac{\|\Pi_\perp \theta\|_1}{\|\theta\|_2}$ and $b(Z_1, Z_2) = \operatorname{sgn}\left(Z_1 \|\widehat{\Pi}_\| \theta^\star\|_2 + Z_2 \|\widehat{\Pi}_\perp \theta^\star\|_2\right) =: \operatorname{sgn}(\beta^\top Z)$ with $\beta^\top = (\|\widehat{\Pi}_\| \theta^\star\|_2, \|\widehat{\Pi}_\perp \theta^\star\|_2)$.

Define the event $A = \{\operatorname{sgn}\left(Z_1 \|\widehat{\Pi}_\| \theta^\star\|_2 + Z_2 \|\widehat{\Pi}_\perp \theta^\star\|_2\right) - \epsilon \frac{\|\Pi_\perp \theta\|_1}{\|\theta\|_2} \leq 0\}$. Because $Z_2$ is symmetric, the distribution of $Z_1 b(Z_1, Z_2)$ is symmetric, hence we can rewrite the risk

$$\mathbf{R}_\epsilon(\theta) = \underbrace{\mathbb{P}(b(Z_1, Z_2) \leq 0 | Z_1 \geq 0)}_{T_1} + \underbrace{\mathbb{P}(Z_1 \leq \nu, b(Z_1, Z_2) \geq 0 | Z_1 \geq 0)}_{T_2} \qquad (17)$$

and derive expression for $T_1, T_2$ separately.

**Step 1: Proof for $T_1$**   Note that due to $\|\theta^\star\|_2 = 1$ we have $\|\beta\|_2 = 1$ and recall that $T_1 = \mathbb{P}(\beta^\top Z \leq 0 | Z_1 \geq 0)$. Using the fact that both $Z_1$ and $Z_2$ are independent standard normal distributed random variables, a simple geometric argument then yields that $T_1 = \frac{\alpha}{\pi}$ with $\alpha = \arccos\left(\frac{\beta_1}{\sqrt{\beta_1^2 + \beta_2^2}}\right) = \arccos(\beta_1)$. Noting that $\beta_1 = \|\widehat{\Pi}_\| \theta^\star\|_2 = \frac{\langle \theta^\star, \theta \rangle}{\|\theta\|_2}$ then yields $T_1 = \frac{1}{\pi} \arccos\left(\frac{\langle \theta^\star, \theta \rangle}{\|\theta\|_2}\right)$.

**Step 2: Proof for $T_2$**   First, assume that $\langle \theta^\star, \theta \rangle \geq 0$. We separate the event $\mathcal{V} = \{Z_1 \leq \nu, b(Z_1, Z_2) \geq 0\}$ into two events $\mathcal{V} = \mathcal{V}_1 \cup \mathcal{V}_2$

$$\mathcal{V}_1 = \{Z_1 \leq \nu, Z_2 \geq 0\} \text{ and } \mathcal{V}_2 = \{Z_1 \leq \nu, b(Z_1, Z_2) \geq 0, Z_2 \leq 0\}.$$

The conditional probability of the first event is directly given

$$\mathbb{P}(\mathcal{V}_1 | Z_1 \geq 0) = \mathbb{P}(Z_2 \geq 0)\mathbb{P}(Z_1 \leq \nu | Z_1 \geq 0) = \frac{1}{2}\operatorname{erf}\left(\frac{\nu}{\sqrt{2}}\right).$$

Hence it only remains to find an expression for $\mathbb{P}(\mathcal{V}_2 | Z_1 \geq 0)$. Letting $\mu$ denote the standard normal distribution, we can write

$$\mathbb{P}(Z_1 \leq \nu, Z_2 \leq 0, b(Z_1, Z_2) \geq 0 | Z_1 \geq 0) = 2\int_0^\nu \int_0^{\frac{\beta_1 x}{\beta_2}} d\mu(y)d\mu(x) = \int_0^\nu \frac{1}{2}\operatorname{erf}\left(\frac{\beta_1 x}{\beta_2}\right) d\mu(x).$$

Together with Step 1, Equation (14) follows by noting that $\beta_1^2 + \beta_2^2 = 1$. Finally, the proof for the case where $\langle \theta^\star, \theta \rangle \leq 0$ follows exactly from the same argument and the proof is complete.   $\square$

### A.3   Distribution shift robustness and consistent adversarial robustness

In this section we introduce distribution shift robustness and show the relation to consistent $\ell_p$-adversarial robustness for certain types of distribution shifts.

When learned models are deployed in the wild, test and train distribution might not be be the same. That is, the test loss might be evaluated on samples from a slightly different distribution than used to train the method. Shifts in the mean of the covariate distribution is a standard intervention studied in the invariant causal prediction literature [10, 11]. For mean shifts in the null space of the ground truth $\theta^\star$ we define an alternative evaluation metric that we refer to as the *distributionally robust risk* defined as follows:

$$\tilde{\mathbf{R}}_\epsilon(\theta) := \max_{\mathbb{Q} \in \mathcal{V}_q(\epsilon; \mathbb{P})} \mathbb{E}_{X \sim \mathbb{Q}} \ell_{\text{test}}(\langle \theta, X + \delta \rangle, \langle \theta^\star, X \rangle), \;\; \text{with}$$

$$\mathcal{V}_p(\epsilon; \mathbb{P}) := \{\mathbb{Q} \in \mathcal{P} : \|\mu_\mathbb{P} - \mu_\mathbb{Q}\|_p \leq \epsilon \text{ and } \langle \mu_\mathbb{P} - \mu_\mathbb{Q}, \theta^\star \rangle = 0\},$$

where $\mathcal{V}_p$ is the neighborhood of mean shifted probability distributions.

A duality between distribution shift robustness and adversarial robustness has been established in earlier work such as [48] for general convex, continuous losses $\ell_{\text{test}}$. For our setting, the following lemma holds.

**Lemma A.3.** *For any $\epsilon \geq 0$ and $\theta$, we have $\tilde{\mathbf{R}}_\epsilon(\theta) \leq \mathbf{R}_\epsilon(\theta)$.*

*Proof.* The proof follows directly from the definition and consistency of the perturbations $\mathcal{U}_p(\epsilon)$ and orthogonality of the mean shifts for the neighborhood $\mathcal{V}_p$. By defining the random variable $W = X - \mu_\mathbb{P}$ for $X \sim \mathbb{P}$ we have the distributional equivalence

$$X' = \mu_\mathbb{P} + \delta + W \overset{d}{=} x + \delta$$

for $X' \sim \mathbb{Q}$ and $X \sim \mathbb{P}$ with $\mu_\mathbb{Q} - \mu_\mathbb{P} = \delta$ and hence

$$\tilde{\mathbf{R}}_\epsilon(\theta) = \max_{\mathbb{Q} \in \mathcal{V}_p(\epsilon)} \mathbb{E}_{X \sim \mathbb{Q}} \ell_{\text{test}}(\langle \theta, X \rangle, \langle \theta^\star, X \rangle) = \max_{\|\delta\|_p \leq \epsilon, \delta \perp \theta^\star} \mathbb{E}_{X \sim \mathbb{P}} \ell_{\text{test}}(\langle \theta, X + \delta \rangle, \langle \theta^\star, X \rangle)$$

$$\leq \mathbb{E}_{X \sim \mathbb{P}} \max_{\|\delta\|_p \leq \epsilon, \delta \perp \theta^\star} \ell_{\text{test}}(\langle \theta, X + \delta \rangle, \langle \theta^\star, X \rangle) = \mathbf{R}_\epsilon(\theta)$$

where the first line follows from orthogonality of the mean-shift to $\theta^\star$. $\square$

# B   Experimental details

In this section we provide additional details on our experiments. All our code including instructions and hyperparameters can be found here: `https://github.com/michaelaerni/interpolation_robustness`.

## B.1   Neural networks on sanitized binary MNIST

Figure 1a shows that robust overfitting in the overparameterized regime also occurs for single hidden layer neural networks on an image dataset that we chose to be arguably devoid of noise. We consider binary classification of MNIST classes 1 vs 3 and further reduce variance by removing "difficult" samples. More precisely, we train networks of width $p \in \{10^1, 10^2, 10^3\}$ on the full MNIST training data and discard all images that take at least one of the models more than 100 epochs of training to fit. While some recent work argues that such sanitation procedures can effectively mitigate robust overfitting [16], we still observe a significant gap between the best (early-stopped) and final test robust accuracies in Figure 1a.

We train all networks on a subset of $n = 2 \times 10^3$ samples using plain mini-batch stochastic gradient descent with learning rate $\nu_p = \sqrt{0.1/p}$ that we multiply by 0.1 after 300 epochs. This learning rate schedule minimizes the training loss efficiently; we did not perform tuning using test or validation data. For the robust test error, we approximate worst-case $\ell_\infty$-perturbations using 10-step SGD attacks on each test sample.

## B.2   Linear and logistic regression

If not mentioned otherwise, we use noiseless i.i.d. samples from our synthetic data model as described in Section 2.1 for our empirical simulations. We calculate all risks in closed-form without noise

and, in the robust case, with consistent perturbations. However, we approximate the integral for the robust 0-1 risk in Theorem 4.1 using a numerical integral solver since we cannot obtain a solution analytically.

For linear regression, we always sample a training set of size $n = 10^3$ and run zero-initialized gradient descent for $2 \times 10^3$ iterations. The learning rate depends on the data dimension $d$ as $\nu_d = \sqrt{1/d}$. Since we observed the training to be initially unstable for large overparameterization ratios $d/n$, we linearly increase the learning rate from zero during the first 250 gradient descent iterations. For evaluation, the linear regression robust population risk always uses consistent $\ell_2$-perturbations of radius $\epsilon = 0.4$. For the noisy case in Figure 1b we set $\sigma^2 = 0.2$.

We fit all logistic regression models except in Figure 3b by minimizing the (regularized) logistic loss from Equation (6) using *CVXPY* in combination with the *Mosek* convex programming solver. Whenever the max-$\ell_2$-margin solution is feasible for $\lambda \to 0$, the problem in Equation (6) has many optimal solutions. In that case, we directly optimize the constrained problem from Equation (7) instead. For Figure 3b, we run zero-initialized gradient descent on the unregularized loss ($\lambda = 0$) for $5 \times 10^5$ iterations. We start with a small initial step size of $0.01$ that we double every $3 \times 10^4$ steps until iteration $3 \times 10^5$. Next, we perform all simulations in Figure 5 on $n = 10^3$ samples from our data model with $d = 8 \times 10^3$. Both training and evaluation use consistent $\ell_\infty$-perturbations of radius $\epsilon = 0.1$. Lastly, for the noisy case in Figure 1c, we flip 2% of all training sample labels.

### B.3 Theoretical predictions

In order to obtain the asymptotic theoretical predictions for logistic regression in Figure 4b corresponding to the empirical simulations with $n = 10^3$ and $\epsilon = 0.05$, we obtain the solution of the optimization problems in Theorem F.1,F.2 with $\epsilon_0 = 0.05\sqrt{10^3\gamma}$ by solving the system of equations $\nabla C = 0$ (with $C$ the optimization objective form Theorem F.1,F.2) using *MATLAB*'s optimization toolbox where we approximate expectations via numerical integration. We note that the optimization problem is numerically challenging to solve, in particular for small values of $\gamma$.

## C  Linear regression – additional insights

In Appendix C.1 we give an intuitive explantion for the robust overfitting phenomenon described in Section 3. Furthermore, in Appendix C.2 we discuss how inconsistent adversarial training prevents interpolation for linear regression.

### C.1  Intuitive explanation

We now shed light on the phenomena revealed by Theorem 3.1 and Figure 2. In particular, we discuss why regularization can reduce the robust risk even in a noiseless setting and why the effect is indiscernible for the standard risk.

For this purpose, we examine the robust risk as a function of $\lambda$, depicted in Figure 7a for different overparameterization ratios $\gamma > 1$ and $\epsilon = 0.4$. The arrows point in the direction of increasing $\lambda$. We observe how the minimal robust risk is achieved for a $\lambda_{\mathrm{opt}}$ bounded away from zero and how the optimum increases with the overparameterization ratio $d/n \to \gamma$, indicating that stronger regularization is needed the more overparameterized the estimator is.

In order to understand this overfitting phenomenon better, we decompose the ridge estimate $\hat{\theta}_\lambda$ into its projection $\Pi_\parallel$ onto the ground truth direction $\theta^\star$ and the projection $\Pi_\perp$ onto the orthogonal complement, i.e., $\hat{\theta}_\lambda = \Pi_\parallel \hat{\theta}_\lambda + \Pi_\perp \hat{\theta}_\lambda$. For the noiseless setting ($\sigma^2 = 0$), substituting this decomposition into Equation (2) yields the following closed-form expression of the robust risk which now involves the parallel error $\|\theta^\star - \Pi_\parallel \hat{\theta}_\lambda\|_2^2$ and the orthogonal error $\|\Pi_\perp \hat{\theta}_\lambda\|_2^2$:

$$\mathbf{R}_\epsilon(\hat{\theta}_\lambda) = \|\theta^\star - \Pi_\parallel \hat{\theta}_\lambda\|_2^2 + (1 + \epsilon^2)\|\Pi_\perp \hat{\theta}_\lambda\|_2^2 + \sqrt{\frac{8\epsilon^2}{\pi}\|\Pi_\perp \hat{\theta}_\lambda\|_2^2(\|\theta^\star - \Pi_\parallel \hat{\theta}_\lambda\|_2^2 + \|\Pi_\perp \hat{\theta}_\lambda\|_2^2)}.$$

(18)

We provide the proof in Appendix A.1.

Figure 7b shows that, as $\lambda$ increases, the orthogonal error decreases faster than the parallel error increases. Since, by Equation (18), the orthogonal error is weighted more heavily for large enough

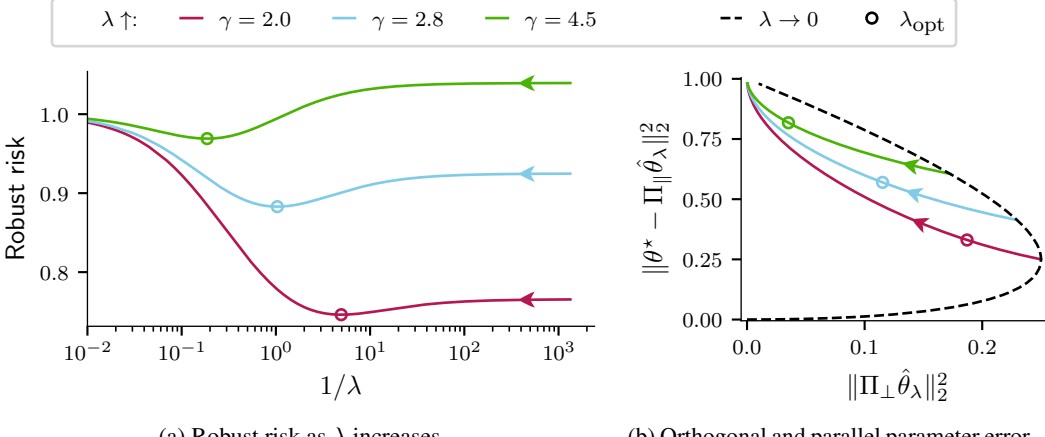

(a) Robust risk as $\lambda$ increases

(b) Orthogonal and parallel parameter error

Figure 7: Theoretical curves depicting the robust risk with $\epsilon = 0.4$ (a) and decomposed terms (b) as $\lambda$ increases (arrow direction) for different choices of the overparameterization ratio $d/n \to \gamma$. In (b) we observe that for large $\gamma > 1$, as $\lambda$ increases, the orthogonal error $\|\Pi_\perp \hat{\theta}_\lambda\|_2$ decreases whereas the parallel error $\|\theta^\star - \Pi_\| \hat{\theta}_\lambda\|_2$ increases. For $\epsilon > 0$, the optimal $\lambda$ is large enough to prevent interpolation.

perturbation strengths $\epsilon$, some nonzero ridge coefficient yields the best trade-off. On the other hand, the standard risk with $\epsilon = 0$ weighs both errors equally, resulting in an optimum at $\lambda \to 0$.

## C.2 Inconsistent adversarial training

As shown in [25] and using the same arguments as in Section A.1, the robust square loss under inconsistent $\ell_2$-perturbations can be reformulated as

$$
\mathcal{L}_\epsilon(\theta) = \frac{1}{n} \sum_{i=1}^{n} (|y_i - \langle x_i, \theta \rangle| + \epsilon \|\theta\|_2)^2
$$

$$
= \frac{1}{n} \sum_{i=1}^{n} (y_i - \langle x_i, \theta \rangle)^2 + \epsilon^2 \|\theta\|_2^2 + \frac{2\epsilon}{n} \|\theta\|_2 \sum_{i=1}^{n} |y_i - \langle x_i, \theta \rangle|.
$$

Thus, we can see that adversarial training with inconsistent perturbations prevents interpolation even when $d > n$, that is, $\mathcal{L}_\epsilon(\theta) = 0$ is unattainable for any $\epsilon > 0$. Nevertheless, we note that optimizing the reformulated robust square loss is equivalent to $\ell_2$-regularized linear regression with $\lambda = \epsilon^2$ and an additional term involving both the weight norm and absolute prediction residuals. We can observe this effect in Figure 4 of [25] where larger $\epsilon$ yield similar effects to larger ridge penalties $\lambda$.

# D  Logistic regression – additional insights

In this section we further discuss robust logistic regression studied in Section 4. Appendix D.1 presents further experiments to contrast consistent and inconsistent perturbations for adversarial training. Furthermore, for completeness, we investigate standard training (that is, $\epsilon = 0$) in Section D.2 and note that it yields significantly worse standard and robust prediction performance.

## D.1 Inconsistent adversarial training

As observed in Section 4.4, label noise can prevent interpolation and hence improve the robust risk of an unregularized estimator with $\lambda \to 0$. We now show similar empirical effects of inconsistent training perturbations with large enough radius $\epsilon$.

Concretely, we perform unregularized ($\lambda \to 0$) adversarial training using consistent vs. inconsistent $\ell_\infty$-perturbations for different $\epsilon$ on fixed $n = 200, 10^3$ and $d = 500$. Figure 8a displays the robust

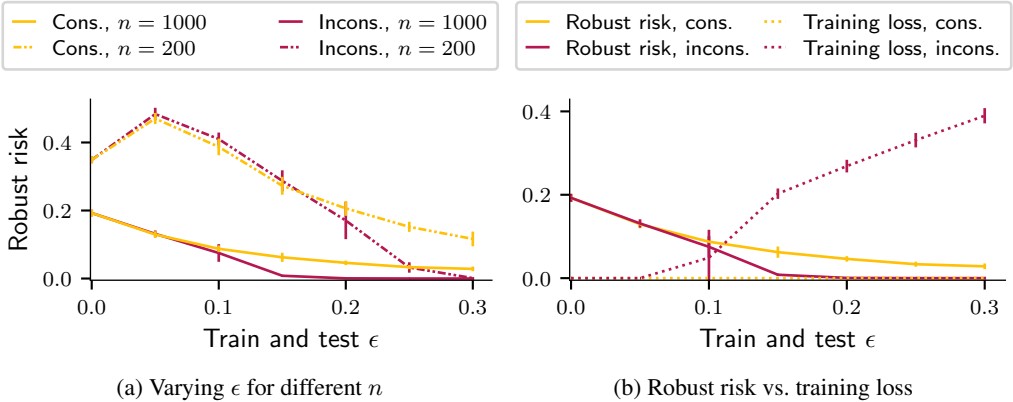

(a) Varying $\epsilon$ for different $n$          (b) Robust risk vs. training loss

Figure 8: Comparison of logistic regression adversarial training with consistent vs. inconsistent $\ell_\infty$-perturbations. (a) Robust risks of the unregularized estimators ($\lambda \to 0$) for $n = 200, 1000$ as $\epsilon$ increases. While for small $\epsilon$, consistent and inconsistent perturbations yield similar robust risks, inconsistent perturbations with large $\epsilon$ outperform consistent perturbations in terms of robustness. We provide an explanation in (b) where we focus on $n = 1000$. In contrast to training with consistent perturbations, inconsistent perturbations may prevent the training loss from vanishing as $\epsilon$ grows large enough. Hence, inconsistent training perturbations can induce spurious regularization effects. We average all experiments over five independent dataset draws from our data model with fixed $d = 500$ and indicate one standard deviation via error bars.

risks of the resulting estimators. For small $\epsilon$, all risks behave very similarly, further corroborating our observations in Figure 4a. However, as the perturbation radius $\epsilon$ grows large, inconsistent perturbations for unregularized adversarial training yield estimators with better robust risk compared to consistent perturbations.

In order to understand this phenomenon, we focus on $n = 10^3$ and depict the robust risk as well as the robust (unregularized) logistic training loss in Figure 8b. We observe that, for large $\epsilon$, inconsistent adversarial training fails to achieve a vanishing loss. Hence, large enough inconsistent perturbations induce noise which prevents interpolation. This observation is similar to the observation made in Section 4.4, where explicit label noise can have spurious regularization effects and in turn, lead to a lower robust risk.

### D.2 Standard vs. adversarial training

Throughout this paper, we focus on adversarial training for logistic regression. For completeness, we also provide simulation results for standard training ($\epsilon = 0$) in Figure 9a. We again use a dataset of size $n = 10^3$. In contrast to adversarial training with $\epsilon > 0$, we do not observe overfitting for neither the standard nor robust risk. However, for $d > n$, the robust risk exhibits its maximum possible value and hence fails to provide any insights. We note that our observations are consistent with [46].

### D.3 Adversarial training with $\ell_2$-perturbations

As mentioned in Section 4, we focus on $\ell_\infty$-perturbations in the context of logistic regression but for completeness also discuss $\ell_2$-perturbations. Following the same argument as in Lemma A.2, it is trivial to see that $\ell_2$-perturbations punish the $\ell_2$-norm of the estimator. Intuitively, we therefore expect that adversarial training with $\ell_2$-perturbations results in an estimator $\hat{\theta}_\lambda$ that is close to a rescaled version of the estimator obtained if training without adversarial perturbations. Since both the robust and standard risk are independent of the estimator scale, we hence do not expect any benefits from explicit $\ell_2$-regularization, i.e., no robust overfitting. Indeed, our simulation results in Figure 9b show almost no regularization benefits for neither the standard nor robust risk.

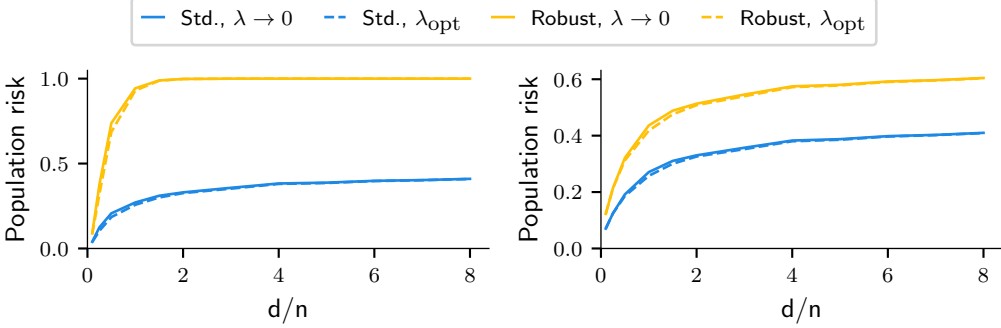

(a) Standard training and $\ell_\infty$-perturbations

(b) Adversarial training with $\ell_2$-perturbations

Figure 9: Additional logistic regression simulations using $n = 10^3$ training samples from our data model for varying degrees of overparameterization $d/n$. (a) Standard training evaluated using consistent $\ell_\infty$-perturbations of radius $\epsilon = 0.1$. (b) Adversarial training using inconsistent $\ell_2$-perturbations of radius $\epsilon = 0.5$ for training and the corresponding consistent perturbation set for evaluation. In both settings, the robust risks are large and not even an optimally weighted ridge penalty helps to reduce them.

## E Proof of Theorem 3.1

In this section, we provide a proof of Theorem 3.1, which characterizes the asymptotic risk of the linear regression estimator $\hat{\theta}_\lambda$ defined in Equation (3).

We first introduce some notation and give the standard closed form solution for the ridge regression estimate $\hat{\theta}_\lambda$. Denoting the input data matrix by $X \in \mathbb{R}^{d \times n}$, the observation vector $y \in \mathbb{R}^n$ reads $y = X^\top \theta^\star + \xi$ where $\xi \sim \mathcal{N}(0, I)$ is the noise vector. The noise vector contains i.i.d. zero-mean $\sigma^2$-variance Gaussian noise as entries. Defining the empirical covariance matrix as $\hat{\Sigma} = \frac{1}{n} X^\top X$ yields the ridge estimate

$$\begin{aligned}
\hat{\theta}_\lambda &= \frac{1}{n}(\lambda I_d + \hat{\Sigma})^{-1} X^\top y \\
&= (\lambda I_d + \hat{\Sigma})^{-1} \hat{\Sigma} \theta^\star + \frac{1}{n}(\lambda I_d + \hat{\Sigma})^{-1} X^\top \xi.
\end{aligned} \tag{19}$$

For $\lambda \to 0$, we obtain the min-norm interpolator

$$\hat{\theta}_0 = \lim_{\lambda \to 0} \hat{\theta}_\lambda = \hat{\Sigma}^\dagger X^\top y,$$

where $\hat{\Sigma}^\dagger$ denotes the Moore-Penrose pseudo inverse.

We now compute the adversarial risk of this estimator. By Equation (11), the adversarial risk depends on the estimator only via the two terms $\|\hat{\theta}_\lambda - \theta^\star\|_2$ and $\|\Pi_\perp \hat{\theta}_\lambda\|_2$. To characterize the asymptotic risk, we hence separately derive asymptotic expressions for each of both terms. The following convergence results hold almost surely with respect to the draws of the train dataset, with input features $X$ and observations $y$, as $n, d \to \infty$.

**Step 1: Characterizing $\|\hat{\theta}_\lambda - \theta^\star\|_2^2$.** Here, we show that

$$\|\hat{\theta}_\lambda - \theta^\star\|_2^2 \to \mathcal{R}_\lambda = \mathcal{B} + \mathcal{V}, \tag{20}$$

where $\mathcal{B} = \lambda^2 m'(-\lambda)$ and $\mathcal{V} = \sigma^2 \gamma(m(-\lambda) - \lambda m'(-\lambda))$ are the asymptotic bias and variance. Hastie et al. [23] considers a similar setup and Theorem 5 of [23] show that $\mathbb{E}_\xi \|\hat{\theta}_\lambda - \theta^\star\|_2^2 \to \mathcal{B} + \mathcal{V}$ and the expectation is taken over the observation noise $\xi$ in the train dataset. In this paper, we define the population risks without the expectation over the noise. Hence, in a first step, the goal is to extend Theorem 5 [23] for the standard risk $\mathbf{R}(\hat{\theta}_\lambda) = \|\hat{\theta}_\lambda - \theta^\star\|_2^2$ such that (20) holds almost surely over the draws of the training data.

Using Equation (19) we can rewrite

$$\|\hat{\theta}_\lambda - \theta^\star\|_2^2 = \| \left( I_d - (\lambda I_d + \widehat{\Sigma})^{-1}\widehat{\Sigma} \right) \theta^\star + \frac{1}{n}(\lambda I_d + \widehat{\Sigma})^{-1}\mathrm{X}^\top\xi\|_2^2$$

$$= \underbrace{\| \left( I_d - (\lambda I_d + \widehat{\Sigma})^{-1}\widehat{\Sigma} \right) \theta^\star\|_2^2}_{T_1} + \underbrace{\langle \frac{\xi}{\sqrt{n}}, (\lambda I_d + \widehat{\Sigma})^{-2}\widehat{\Sigma}\frac{\xi}{\sqrt{n}}\rangle}_{T_2}$$

$$+ \underbrace{\left\langle \frac{\mathrm{X}^\top}{\sqrt{n}}(\lambda I_d + \widehat{\Sigma})^{-1} \left( I_d - (\lambda I_d + \widehat{\Sigma})^{-1}\widehat{\Sigma} \right) \theta^\star, \frac{\xi}{\sqrt{n}} \right\rangle}_{T_3},$$

where we used for the second equality that $\langle \frac{\xi}{\sqrt{n}}, \frac{\mathrm{X}}{\sqrt{n}}(\lambda I_d + \widehat{\Sigma})^{-2}\frac{\mathrm{X}^\top}{\sqrt{n}}\frac{\xi}{\sqrt{n}}\rangle = \langle \frac{\xi}{\sqrt{n}}, (\lambda I_d + \widehat{\Sigma})^{-2}\widehat{\Sigma}\frac{\xi}{\sqrt{n}}\rangle$.

The first term $T_1 \to \mathcal{B}$ follows directly via Theorem 5 [23]. We next show that $T_2 \to \mathcal{V}$ and $T_3 \to 0$ almost surely, which establishes Equation 20.

**Proof that $T_2 \to \mathcal{V}$:** While Theorem 5 [23] also shows that $\mathbb{E}_\xi \operatorname{tr} \left( \frac{1}{n}\xi\xi^\top\widehat{\Sigma}(\lambda I_d + \widehat{\Sigma})^{-2} \right) \to \mathcal{V}$, we require the convergence almost surely over a single draw of $\xi$. In fact, this directly follows from the same argument as used for the proof of Theorem 5 [23] and the fact that $\|\frac{\xi}{\sqrt{n}}\|_2^2 \to \sigma^2$. Hence $\langle \frac{\xi}{\sqrt{n}}, (\lambda I_d + \widehat{\Sigma})^{-2}\widehat{\Sigma}\frac{\xi}{\sqrt{n}}\rangle \to \mathcal{V}$ almost surely over the draws of $\xi$.

**Proof that $T_3 \to 0$:** This follows straight forwardly from sub-Gaussian concentration inequalities and from the fact that

$$\left\| \frac{\mathrm{X}}{\sqrt{n}}(\lambda I_d + \widehat{\Sigma})^{-1} \left( I_d - (\lambda I_d + \widehat{\Sigma})^{-1}\widehat{\Sigma} \right) \theta^\star \right\|_2 = O(1),$$

which is a direct consequence of the Bai-Yin theorem [4], stating that for sufficiently large $n$, the non zero eigenvalues of $\widehat{\Sigma}$ can be almost surely bounded by $(1 + \sqrt{\gamma})^2 \geq \lambda_{\max}(\widehat{\Sigma}) \geq \lambda_{\min}(\widehat{\Sigma}) \geq (1 - \sqrt{\gamma})^2$. Hence we can conclude the first part of the proof.

**Step 2: Characterizing $\|\Pi_\perp\hat{\theta}_\lambda\|_2$.** Here, we show that

$$\|\Pi_\perp\hat{\theta}_\lambda\|_2^2 \to \mathcal{R}_\lambda - \lambda^2(m(-\lambda))^2 =: \mathcal{P}_\lambda. \tag{21}$$

We assume without loss of generality that $\|\theta^\star\|_2 = 1$ and hence $\Pi_\perp = I_d - \theta^\star(\theta^\star)^\top$. It follows that

$$\|\Pi_\perp\hat{\theta}_\lambda\|_2^2 = \|\hat{\theta}_\lambda\|_2^2 - \left( \langle\hat{\theta}_\lambda, \theta^\star\rangle \right)^2$$

$$= \|\theta^\star - \hat{\theta}_\lambda - \theta^\star\|_2^2 - \left( 1 - \langle\theta^\star - \hat{\theta}_\lambda, \theta^\star\rangle \right)^2$$

$$= \|\theta^\star - \hat{\theta}_\lambda\|_2^2 - 2\langle\theta^\star - \hat{\theta}_\lambda, \theta^\star\rangle + 1 - \left( 1 - \langle\theta^\star - \hat{\theta}_\lambda, \theta^\star\rangle \right)^2$$

$$= \|\theta^\star - \hat{\theta}_\lambda\|_2^2 - \left( \langle\theta^\star - \hat{\theta}_\lambda, \theta^\star\rangle \right)^2.$$

The convergence of the first term is already known form step 1. Hence, it is only left to find an asymptotic expression for $\langle\theta^\star - \hat{\theta}_\lambda, \theta^\star\rangle$. Inserting the closed form expression from Equation (19), we obtain:

$$\langle\theta^\star - \hat{\theta}_\lambda, \theta^\star\rangle = \langle I_d - \left( \lambda I_d + \widehat{\Sigma} \right)^{-1}\widehat{\Sigma} \rangle \theta^\star, \theta^\star\rangle - \langle(\lambda I_d + \widehat{\Sigma})^{-1}\frac{\mathrm{X}^\top\xi}{n}, \theta^\star\rangle. \tag{22}$$

Note that $\langle(\lambda I_d + \widehat{\Sigma})^{-1}\frac{\mathrm{X}\xi}{n}, \theta^\star\rangle$ vanishes almost surely over the draws of $\xi$ using the same reasoning as in the first step. Hence, we only need to find an expression for the first term on the RHS of Equation (22). Note that we can use Woodbury's matrix identity to write:

$$\langle I_d - \left( \lambda I_d + \widehat{\Sigma} \right)^{-1}\widehat{\Sigma} \rangle \theta^\star, \theta^\star\rangle = \lambda\langle(\lambda I_d + \widehat{\Sigma})^{-1}\theta^\star, \theta^\star\rangle.$$

However, the expression on the RHS appears exactly in the proof of Theorem 1 [23] (Equation 116), which shows that $\lambda\langle(\lambda I_d + \widehat{\Sigma})^{-1}\theta^\star, \theta^\star\rangle \to \lambda m(-\lambda)$ with $m(z)$ as in Theorem 3.1. Hence the proof of almost sure convergence (21) of $\|\Pi_\perp\hat{\theta}_\lambda\|_2$ is complete.

Substituting Equations (20) and (21) into robust risk (11) expression yields:

$$\mathbf{R}_\epsilon(\hat{\theta}_\lambda) \xrightarrow{\text{a.s.}} \mathcal{R}_\lambda + \epsilon^2\mathcal{P}_\lambda + \sqrt{\frac{8\epsilon^2}{\pi}\mathcal{P}_\lambda\mathcal{R}_\lambda} = \mathcal{R}_{\epsilon,\lambda}.$$

Finally, we note that $\lim_{\lambda\to0}\mathcal{R}_{\epsilon,\lambda}$ exists and is finite for any $\gamma \neq 1$ since: $\lim_{\lambda\to0} m(-z) = \frac{1}{1-\gamma}$ for $\gamma < 1$ and $\lim_{\lambda\to0} m(-z) = \frac{1}{\gamma(\gamma-1)}$ for $\gamma > 1$, $\lim_{z\to0} zm'(-z) = 0$, $\lim_{z\to0} z^2m'(-z) = 0$ for $\gamma < 1$ and $\lim_{z\to0} z^2m'(-z) = 1 - \frac{1}{\gamma}$ for $\gamma > 1$ (see also Corollary 5 in [23]). Hence, we can conclude from the continuity of the risk that $\mathbf{R}_\epsilon(\hat{\theta}_0) \xrightarrow{\text{a.s.}} \lim_{\lambda\to0}\mathcal{R}_{\epsilon,\lambda}$ and therefore, the proof is complete.

# F    Details on Theorem 4.1

In this section we give a formal statement for Theorem 4.1. The results are based on the Convex Gaussian Minimax Theorem (CGMT) [20, 51]. We first prove the case when training with consistent perturbations (9) and noiseless observations. Then, we show how the theorems extend to the case when training with inconsistent perturbations (8) and training label noise.

The results presented in this section have similarities with the ones in [24]. However, we study a discriminative data model with features drawn from a single Gaussian and a 1-sparse ground truth. In contrast, the authors of [24] study a generative data model with features drawn from two Gaussians. Furthermore, several papers study logistic regression for isotropic Gaussian features in high dimensions [46, 50], but focus their analysis on the standard risk and do not consider adversarial robustness.

An immediate consequence of the proof of Lemma A.2 is that the adversarial loss from Equation (6) with respect to consistent $\ell_\infty$-attacks (9) for the 1-sparse ground truth has the closed-form equivalent

$$\mathcal{L}_{\epsilon,\lambda}(\theta) = \frac{1}{n}\sum_{i=1}^{n}\ell_{\text{train}}(y_i\langle\theta, x_i\rangle - \epsilon\|\Pi_\perp\hat{\theta}\|_1) + \lambda\|\theta\|_2^2, \tag{23}$$

where $\Pi_\perp$ is the projection matrix to the orthogonal subspace of $\theta^\star$.

Let $\mathcal{M}_f(x,t) = \min_y \frac{1}{2t}(x-y)^2 + f(y)$ be the Moreau envelope and let $Z_\parallel, Z_\perp$ be two independent standard normal random variables. We can now state Theorem F.1 that describes the asymptotic risk of $\hat{\theta}_\lambda(\epsilon)$, for $\lambda > 0$ and for the asymptotic regime where $d, n \to \infty$. The proof of the theorem can be found in Appendix F.1.

**Theorem F.1.** *Assume that we have i.i.d. random features $x_i$ drawn from an isotropic Gaussian, noiseless observations $y_i = \text{sgn}(\langle x_i, \theta^\star\rangle)$, and ground truth $\theta^\star = (1, 0, \ldots, 0)^\top$. Further, assume that $\lambda > 0$ and $\epsilon = \epsilon_0/\sqrt{d}$, where $\epsilon_0$ is a numerical constant. Let $(\nu_\perp^\star, \nu_\parallel^\star, r^\star, \delta^\star, \mu^\star, \tau^\star)$ be the unique solution of*

$$\min_{\substack{\nu_\perp\geq0,\tau\geq0,\\ \nu_\parallel,\delta\geq0}} \max_{\substack{r\geq0,\\ \mu\geq0}} \mathbb{E}_{Z_\parallel,Z_\perp}\left[\mathcal{M}_\ell(|Z_\parallel|\nu_\parallel + Z_\perp\nu_\perp - \epsilon_0\delta, \frac{\tau}{r})\right] - \delta\mu + \frac{r\tau}{2} + \lambda(\nu_\perp^2 + \nu_\parallel^2)$$

$$-\nu_\perp\sqrt{\left[(\mu^2 + \gamma r^2) - (\mu^2 + \gamma r^2)\text{erf}(\mu/(\sqrt{\gamma}r\sqrt{2})) - \sqrt{\frac{2}{\pi}}\sqrt{\gamma}r\mu\exp(-\mu^2/(\gamma r^2 2))\right]}. \tag{24}$$

*Then, for $\lambda > 0$, the estimator $\hat{\theta}_\lambda(\epsilon)$ from Equation (6) with the logistic loss and consistent $\ell_\infty$-perturbations satisfies asymptotically as $d, n \to \infty$ and $d/n \to \gamma$ that*

$$\frac{1}{\sqrt{d}}\|\Pi_\perp\hat{\theta}_\lambda(\epsilon)\|_1 \to \delta^\star \quad \text{and} \quad \langle\hat{\theta}_0(\epsilon), \theta^\star\rangle \to \nu_\parallel^\star \quad \text{and} \quad \|\hat{\theta}_\lambda(\epsilon)\|_2^2 \to \nu_\parallel^{\star2} + \nu_\perp^{\star2}. \tag{25}$$

*The convergences hold in probability.*

For $\lambda > 0$, the loss in Equation (23) has a unique minimizer. In contrast, for $\lambda = 0$, the minimizer of Equation (23) is not unique. In the latter case, we study the robust max-$\ell_2$-margin solution from Equation (7), which corresponds to the limit when $\lambda \to 0$ (see Section 4.1). Theorem F.2 characterizes the asymptotic behavior of the corresponding solution, with proof in Appendix F.2.

**Theorem F.2.** *Assume that we have i.i.d. random features $x_i$ drawn from an isotropic Gaussian, noiseless observations $y_i = \text{sgn}(\langle x_i, \theta^\star \rangle)$, and $\theta^\star = (1, 0, \cdots, 0)^\top$. Further, assume that $\lambda = 0$ and $\epsilon = \epsilon_0/\sqrt{d}$, where $\epsilon_0$ is a numerical constant. Let $(\nu_\perp^\star, \nu_\parallel^\star, r^\star, \delta^\star, \zeta^\star, \kappa^\star, \tau^\star)$ be the unique solution of*

$$
\max_{\substack{r \geq 0,\ \nu_\perp \geq 0, \\ \zeta \geq 0\ \nu_\parallel, \delta \geq 0}} \min \ \max_{\kappa \geq 0} \min_{\tau \geq 0}\ \nu_\parallel^2 - \kappa \nu_\perp - \delta \zeta - \frac{\gamma r^2}{4(1 + \frac{\kappa}{2\tau})}
$$
$$
+ r \sqrt{\mathbb{E}_{Z_\parallel, Z_\perp}\left[ \max\left(0, 1 + \epsilon_0 \delta - |Z_\parallel| \nu_\parallel + Z_\perp \nu_\perp \right)^2 \right]} \tag{26}
$$
$$
+ \frac{1}{2(1 + \frac{\kappa}{2\tau})} \left( \frac{\gamma r^2 + \zeta^2}{2} \text{erf}\left( \frac{\zeta}{\sqrt{2}\sqrt{\gamma} r} \right) - \frac{\zeta^2}{2} + \frac{\sqrt{\gamma} r \zeta}{\sqrt{2\pi}} \exp\left( -\frac{\zeta^2}{2\gamma r^2} \right) \right) + \frac{\kappa \tau}{2}.
$$

*Then, the estimator $\hat{\theta}_0(\epsilon)$ from Equation (7) with the logistic loss and consistent $\ell_\infty$-perturbations satisfies asymptotically as $d, n \to \infty$ and $d/n \to \gamma$ that*

$$
\frac{1}{\sqrt{d}} \|\Pi_\perp \hat{\theta}_0(\epsilon)\|_1 \to \delta^\star \ \text{ and } \ \langle \hat{\theta}_0(\epsilon), \theta^\star \rangle \to \nu_\parallel^\star \ \text{ and } \ \|\hat{\theta}_0(\epsilon)\|_2^2 \to \nu_\parallel^{\star 2} + \nu_\perp^{\star 2}. \tag{27}
$$

*The convergences hold in probability.*

**Remark F.3.** *Theorem 4.1 follows from Theorems F.1 and F.2 when inserting the expression from Equations (25),(27) into the expression of the risk in Lemma A.2.*

**Inconsistent adversarial attacks**   We now show that Theorems F.1,F.2 also hold when training with inconsistent attacks (8).

For inconsistent adversarial attacks, we simply need to change $\epsilon \|\Pi_\perp \theta\|_1$ to $\epsilon \|\theta\|_1 = \epsilon \|\Pi_\perp \theta\|_1 + \epsilon \|\Pi_\parallel \theta\|_1$ in the optimization objective in Equations (28),(38). To show that these modifications do not change the asymptotic solution as $d, n \to \infty$, note that $\epsilon \|\Pi_\parallel \theta\|_1 = \frac{\epsilon_0}{\sqrt{n}} \|\Pi_\parallel \theta\|_1 \to 0$ which follows from the fact that $\|\Pi_\parallel \theta\|_1$ remains bounded as $d, n \to \infty$.

**Label noise**   While our results assume noiseless observations $y_i = \text{sgn}(\langle x_i, \theta^\star \rangle)$, Theorem F.1,F.2 can be extended to the case where additional label noise is added to the observations. That is, we observe $y_i = \text{sgn}(\langle x_i, \theta^\star \rangle)\xi_i$ with $\xi_i$ i.i.d., $\mathbb{P}(\xi_i = 1) = 1 - \sigma$ and $\mathbb{P}(\xi_i = -1) = \sigma$, where $\sigma$ is the strength of the label noise.

Note that, as discussed in Section D.1, the robust max-margin solution (7) might not exist for noisy observations. In that case, the robust logistic regression estimate (6) has a unique solution for $\lambda = 0$. In fact, following the same argument as in [24], asymptotically, we can find a threshold $\gamma^\star$ such that for any $\gamma < \gamma^\star$, the robust max-$\ell_2$-margin solution does not exist, and for any $\gamma \geq \gamma^\star$, the robust max-$\ell_2$-margin solution exists. The threshold can be found using the CGMT when following the same argument as in Theorem 6.1 of [24].

Finally, we remark that, when $\lambda > 0$ or $\lambda = 0$ and $\gamma < \gamma^\star$, we can extend Theorem F.1 by replacing $|Z_\parallel|$ with $\xi |Z_\parallel|$, where $\xi$ is drawn from the same distribution as $\xi_i$ defined above. Similarly, for $\lambda = 0$ and $\gamma \geq \gamma^\star$, we can extend Theorem F.2 by replacing $|Z_\parallel|$ with $\xi |Z_\parallel|$.

**Outline of the proof**   The proof of Theorems F.1,F.2 heavily relies on the proofs of Theorem 6.3 and 6.4 in [24]. In particular, our proof essentially follows the same structure by first reducing the problem via an application of the Lagrange multiplier to an expression that suits the CGMT framework. This allows us to instead study the auxiliary optimization problem as described in Equation (31), which we then simplify to a scalar optimization problem using standard concentration inequalities of Gaussian random variables.

The major difference to Theorems 6.3 and 6.4 in [24] is that we study a discriminative data model with a sparse ground truth, whereas Theorem 6.3 and 6.4 in [24] assume a generative data model and,

in particular, do not allow sparse ground truth vectors $\theta^\star$. This is due to the different attack sizes as we choose $\epsilon = \epsilon_0/\sqrt{d}$ while Theorems 6.3 and 6.4 in [24] assume a constant attack size $\epsilon = \epsilon_0$.

## F.1 Proof of Theorem F.1

Denote with $X \in \mathbb{R}^{n \times d}$ the input data matrix and with $y \in \mathbb{R}^n$ the vector containing the observations. Recall that the estimator $\hat\theta$ is given by

$$\hat\theta = \arg\min_\theta \frac{1}{n} \sum_{i=1}^n \ell(y_i \langle x_i, \theta \rangle - \epsilon \|\Pi_\perp \theta\|_1) + \lambda \|\theta\|_2^2$$

$$= \arg\min_{\theta,v} \frac{1}{n} \sum_{i=1}^n \ell(v_i - \epsilon \|\Pi_\perp \theta\|_1) + \lambda \|\theta\|_2^2 \text{ such that } v = D_y X \theta, \tag{28}$$

where $\ell(x) = \log(1 + \exp(-x))$ is the logistic loss, $X \in \mathbb{R}^{n \times d}$ is the data matrix and $D_y$ the diagonal matrix with entries $(D_y)_{i,i} = y_i$. We can then introduce the Lagrange multipliers $u \in \mathbb{R}^n$ to obtain

$$\min_{\theta,v} \max_u \frac{1}{n} \sum_{i=1}^n \ell(v_i - \epsilon \|\Pi_\perp \theta\|_1) + \frac{1}{n} u^\top D_y X \theta - \frac{1}{n} u^\top v + \lambda \|\theta\|_2^2.$$

Furthermore, we can separate $X = X\Pi_\perp + X\Pi_\|$, which yields

$$\min_{\theta,v} \max_u \frac{1}{n} \sum_{i=1}^n \ell(v_i - \epsilon \|\Pi_\perp \theta\|_1) + \frac{1}{n} u^\top D_y X\Pi_\| \theta + \frac{1}{n} u^\top D_y X\Pi_\perp \theta - \frac{1}{n} u^\top v + \lambda \|\theta\|_2^2. \tag{29}$$

**Convex Gaussian Minimax Theorem** We can now make use of the CGMT, which states that

$$\min_{\theta \in U_\theta} \max_{u \in U_u} u^\top X \theta + \psi(u, \theta), \tag{30}$$

with $\psi$ convex in $\theta$ and concave in $u$, has asymptotically, when $d, n \to \infty$, $d/n \to \gamma$, pointwise the same solution as

$$\min_{\theta \in U_\theta} \max_{u \in U_u} \|u\|_2 g^\top \theta + u^\top h \|\theta\|_2 + \psi(u, \theta), \tag{31}$$

where $g \in \mathbb{R}^d$ and $h \in \mathbb{R}^n$ are random vectors with i.i.d. standard normal entries, and $U_\theta$ and $U_u$ are compact sets. As is common in the literature, we call Equation (30) the primal optimization problem and Equation (31) the auxiliary optimization problem. Several works have already used the CGMT to study high dimensional asymptotic logistic regression [46], also when training with adversarial attacks [24]. We omit the precise statement and refer the reader to [51]. However, we note that we can apply the CGMT due to the following observations:

1. The objective (29) is concave in $u$ and convex in $v, \theta$.
2. We can restrict $u, v, \theta$ to compact sets without changing the solution. For $\theta$, we note that this is a consequence of $\lambda > 0$, and for $u, v$, we note that the stationary condition requires $u_i = \ell'(v_i - \epsilon \|\Pi_\perp \theta\|_1)$.
3. $X\Pi_\perp$ is independent of the observations $y$ and of $X\Pi_\|$.

Therefore, as a consequence of the CGMT, we can show that the solution of the primal optimization problem (29) asymptotically concentrates around the same value as the solution of the following auxiliary optimization problem:

$$\min_{\theta,v} \max_u \frac{1}{n} \sum_{i=1}^n \ell(v_i - \epsilon \|\Pi_\perp \theta\|_1) + \frac{1}{n} u^\top D_y X\Pi_\| \theta + \frac{1}{n} \|u^\top D_y\|_2 g^\top \Pi_\perp \theta$$

$$+ \frac{1}{n} u^\top D_y h \|\Pi_\perp \theta\|_2 - \frac{1}{n} u^\top v + \lambda \|\theta\|_2^2,$$

where $g \in \mathbb{R}^d$ and $h \in \mathbb{R}^n$ are vectors with i.i.d. standard normal entries.

**Scalarization of the optimization problem** We now aim to simplify the optimization problem. In a first step, we maximize over $u$. For this, define $r = \|u\|_2/\sqrt{n}$, which allows us to equivalently write

$$\min_{\theta,v} \max_{r\geq 0} \; \frac{1}{n}\sum_{i=1}^{n}\ell(v_i - \epsilon\|\Pi_\perp\theta\|_1) + \frac{r}{\sqrt{n}}\|D_y\mathrm{X}\Pi_\|\theta + D_y h\|\Pi_\perp\theta\|_2 - v\|_2 + \frac{1}{\sqrt{n}}rg^\top\Pi_\perp\theta + \lambda\|\theta\|_2^2,$$

where we have used the fact that $\|u^\top D_y\|_2 = \|u\|_2$. In order to proceed, we want to separate $\Pi_\perp\theta$ from the loss $\ell(v, \Pi_\perp\theta) := \frac{1}{n}\sum_{i=1}^{n}\ell(v_i - \epsilon\|\Pi_\perp\theta\|_1)$. Denoting the conjugate of $\ell$ by $\tilde{\ell}$, we can write $\ell(v, \Pi_\perp\theta)$ in terms of its conjugate with respect to $\Pi_\perp\theta$:

$$\ell(v, \Pi_\perp\theta) = \sup_{w} \frac{1}{\sqrt{d}} w^\top \Pi_\perp\theta - \tilde{\ell}(v, w)$$

$$= \sup_{w} \frac{1}{\sqrt{d}} w^\top \Pi_\perp\theta - \sup_{\delta\geq 0}\left(\frac{\sqrt{d}}{\sqrt{d}}\delta\|w\|_\infty - \frac{1}{n}\sum_{i=1}^{n}\ell(v_i - \sqrt{d}\epsilon\delta)\right)$$

$$= \sup_{w}\inf_{\delta\geq 0} \frac{1}{\sqrt{d}} w^\top \Pi_\perp\theta - \delta\|w\|_\infty + \frac{1}{n}\sum_{i=1}^{n}\ell(v_i - \epsilon_0\delta),$$

where, for the second identity, we use the derivation for the conjugate of $\ell$ from Lemma A.2 in the paper [24]. Hence, we obtain:

$$\max_{r\geq 0}\min_{\theta,v}\max_{w}\min_{\delta\geq 0} \; \frac{1}{n}\sum_{i=1}^{n}\ell(v_i - \epsilon_0\delta) + \frac{r}{\sqrt{n}}\left\|D_y\mathrm{X}\Pi_\|\theta + D_y h\|\Pi_\perp\theta\|_2 - v\right\|_2 + \lambda\|\theta\|_2^2 \quad (32)$$

$$+ \frac{1}{\sqrt{d}}w^\top\Pi_\perp\theta - \delta\|w\|_\infty + \frac{1}{\sqrt{n}}rg^\top\Pi_\perp\theta. \quad (33)$$

In particular, note that the problem is concave in $r, w$ and convex in $\theta, v, \delta$. Thus, we can interchange the order of maximization and minimization:

$$\max_{r\geq 0}\min_{v}\min_{\delta\geq 0}\max_{w}\min_{\theta} \; \frac{1}{n}\sum_{i=1}^{n}\ell(v_i - \epsilon_0\delta) + \frac{r}{\sqrt{n}}\left\|D_y\mathrm{X}\Pi_\|\theta + D_y h\|\Pi_\perp\theta\|_2 - v\right\|_2 + \lambda\|\theta\|_2^2$$

$$(34)$$

$$+ \frac{1}{\sqrt{d}}w^\top\Pi_\perp\theta - \delta\|w\|_\infty + \frac{1}{\sqrt{n}}rg^\top\Pi_\perp\theta.$$

Next, we simplify the optimization over $\theta$. Write $\Pi_\|\theta = \Pi_\| 1\nu_\|$ with $\nu_\| \in \mathbb{R}$ (here we use the fact that $\theta^\star = (1, 0, \cdots, 0)$) and let $\nu_\perp = \|\Pi_\perp\theta\|_2$. We can simplify:

$$\max_{r\geq 0}\min_{\substack{\nu_\perp\geq 0,\\ \delta\geq 0,\\ \nu_\|,v}}\max_{w} \; \frac{1}{n}\sum_{i=1}^{n}\ell(v_i - \epsilon_0\delta) + \frac{r}{\sqrt{n}}\|D_y\mathrm{X}\Pi_\| 1\nu_\| + D_y h\nu_\perp - v\|_2 + \lambda(\nu_\|^2 + \nu_\perp^2) \quad (35)$$

$$- \frac{1}{\sqrt{d}}\nu_\perp\|\Pi_\perp(w - \sqrt{\gamma}rg)\|_2 - \delta\|w\|_\infty$$

In order to obtain a low dimensional scalar optimization problem, we still need to scalarize the optimization over $w$ and $v$. For this, we replace the term $\|D_y\mathrm{X}\Pi_\| 1\nu_\| + D_y h\nu_\perp - v\|_2$ with its square, which is achieved by using the following identity $\min_{\tau\geq 0}\frac{x^2}{2\tau} + \frac{\tau}{2} = x$. Hence,

$$\max_{r\geq 0}\min_{\substack{\nu_\perp\geq 0,\tau\geq 0,\\ \delta\geq 0,\\ \nu_\|,v}} \; \frac{1}{n}\sum_{i=1}^{n}\ell(v_i - \epsilon_0\delta) + \frac{r}{2\tau n}\|D_y\mathrm{X}\Pi_\| 1\nu_\| + D_y h\nu_\perp - v\|_2^2 + \frac{\tau r}{2} + \lambda(\nu_\|^2 + \nu_\perp^2)$$

$$+ \max_{w}\left[-\frac{1}{\sqrt{d}}\nu_\perp\|\Pi_\perp(w - \sqrt{\gamma}rg)\|_2 - \delta\|w\|_\infty\right].$$

We can now separately solve the following two inner optimization problems:

$$\max_{w} \; -\nu_\perp\frac{1}{d}\|\Pi_\perp(w - \sqrt{\gamma}rg)\|_2 - \delta\|w\|_\infty \quad (36)$$

$$\min_{v} \; \frac{r}{2\tau n}\|D_y\mathrm{X}\Pi_\| 1\nu_\| + D_y h\nu_\perp - v\|_2^2 + \sum_{i=1}^{n}\ell(v_i - \epsilon_0\delta) \quad (37)$$

**Equation (36)** Let $\text{ST}_t(x) = \begin{cases} 0 & |x| \leq t \\ \text{sgn}(x)(|x| - t) & \text{else} \end{cases}$ be the soft threshold function. We have

$$\max_w \ -\nu_\perp \frac{1}{\sqrt{d}} \|\Pi_\perp(w - \sqrt{\gamma}rg)\|_2 - \delta\|w\|_\infty$$

$$= -\min_w \ \nu_\perp \frac{1}{\sqrt{d}}\|\Pi_\perp(w - \sqrt{\gamma}rg)\|_2 + \delta\|w\|_\infty$$

$$\overset{\mu = \|w\|_\infty}{=} -\min_{\mu \geq 0} \nu_\perp \sqrt{\frac{1}{d}\sum_{i=2}^{d}(\text{ST}_\mu(\sqrt{\gamma}rg_i))^2} + \delta\mu$$

$$\overset{\text{LLN as } d \to \infty}{\to} -\min_{\mu \geq 0} \nu_\perp \sqrt{\mathbb{E}_Z(\text{ST}_\mu(\sqrt{\gamma}rZ))^2} + \delta\mu,$$

where we used in the third line that the ground truth $\theta^\star$ is 1-sparse and in the last line that the expectation exists for $Z \sim \mathcal{N}(0,1)$. Finally, we can further simplify

$$\mathbb{E}_Z(\text{ST}_\mu(\sqrt{\gamma}rZ))^2 = \gamma r^2 \mathbb{E}_Z\left(\text{ST}_{\mu/(\sqrt{\gamma}r)}(Z)\right)^2$$

$$= \gamma r^2 \mathbb{E}_Z(Z - \mu/(\sqrt{\gamma}r))^2 - \mathbb{E}_Z \mathbb{1}_{|Z| \leq \mu/(\sqrt{\gamma}r)}(Z - \mu/(\sqrt{\gamma}r))^2$$

$$= (\mu^2 + \gamma r^2)\left(1 - \text{erf}(\mu/(\sqrt{2\gamma}r))\right) - \sqrt{\gamma}r\mu\sqrt{\frac{2}{\pi}}\exp(-\mu/(2\gamma r^2)).$$

Hence, we can conclude the first term.

**Equation (37)** For the second term we also aim to apply the law of large numbers. We have

$$\min_v \ \frac{r}{2\tau n}\|D_y X\Pi_\parallel 1\nu_\parallel + D_y h\nu_\perp - v\|_2^2 + \frac{1}{n}\sum_{i=1}^{n}\ell(v_i - \epsilon_0\delta)$$

$$\overset{\tilde{v} = v - \epsilon\delta}{=} \min_{\tilde{v}} \ \frac{r}{2\tau n}\|D_y X\Pi_\parallel 1\nu_\parallel + D_y h\nu_\perp - \tilde{v} - \epsilon_0\delta\|_2^2 + \frac{1}{n}\sum_{i=1}^{n}\ell(\tilde{v}_i)$$

$$= \min_{\tilde{v}} \ \frac{1}{n}\sum_{i=1}^{n}\frac{r}{2\tau n}\left((D_y X\Pi_\parallel 1\nu_\parallel)_i + (D_y h\nu_\perp)_i - \tilde{v}_i - \epsilon_0\delta\right)^2 + \ell(\tilde{v}_i)$$

$$\overset{\text{LLN}}{\to} \mathbb{E}_{Z_\parallel, Z_\perp}\left[\mathcal{M}_\ell(|Z_\parallel|\nu_\parallel + Z_\perp\nu_\perp - \epsilon_0\delta, \frac{\tau}{r})\right],$$

where in the last line we used that $(D_y X\Pi_\parallel 1)_i = y_i x_i^\top \theta^\star = \text{sgn}(x_i^\top \theta^\star)x_i^\top \theta^\star$ has the same distribution as $|Z_\parallel|$ with $Z_\parallel \sim \mathcal{N}(0,1)$. Further, to apply the law of large numbers, we need to show that the Moreau envelope exists. Similarly to Theorem 1 [46], this follows immediately when noting that $\mathcal{M}_\ell(x, \mu) \leq \ell(x) = \log(1 + \exp(-x)) \leq \log(2) + |x|$. Finally, we obtain the desired optimization problem in Equation (24) when combining these results.

**Convergence** One can check that the optimization problems defined in Equations (24),(28) are convex in the variables that we minimize over, and concave in the variables that we maximize over. Indeed, Equation (28) is immediate and Equation (24) follows straightforwardly from the fact that the problem in Equation (32) is convex and concave as desired. Therefore, also the problem in Equation (24) satisfies the convexity and concavity properties that we need. Hence, both problems in Equations (24),(28) have a unique solution. Finally, note that the optimum $\delta^\star$ in Equation (32) satisfies $\delta^\star = \frac{1}{\sqrt{d}}\|\Pi_\perp \theta\|$, and similarly the optima $\nu_\perp^\star$ and $\nu_\parallel^\star$ in Equation (35) satisfy $\nu_\perp^\star = \|\Pi_\perp \theta\|_2$ and $\nu_\parallel^\star = \langle\theta, \theta^\star\rangle$. We can therefore conclude the proof as the solutions of the optimization problems (24), (28) concentrate asymptotically around the same optima as $d, n \to \infty$.

## F.2 Proof of Theorem F.2

Recall the robust max-margin solution from Equation (7):

$$\min_{\theta, \delta} \|\theta\|_2^2 \text{ such that } \langle\theta, x_i\rangle - \delta \geq 1 \text{ for all } i \text{ and } \epsilon\|\Pi_\perp \theta\|_1 = \delta \tag{38}$$

Like in the previous section, after introducing the Lagrange multipliers $\zeta$ and $u$ we can equivalently write

$$\min_{\theta,\delta} \max_{\substack{u:u_i\geq 0,\\ \zeta\geq 0}} \|\theta\|_2^2 + \frac{1}{n}u^\top\left(1 + 1\epsilon_0\delta - D_yX\theta\right) + \zeta\left(\frac{\|\Pi_\perp\theta\|_1}{\sqrt{d}} - \delta\right), \tag{39}$$

and again separating $X = X\Pi_\perp + X\Pi_\|$, we get

$$\min_{\theta,\delta} \max_{\substack{u:u_i\geq 0,\\ \zeta\geq 0}} \|\theta\|_2^2 + \frac{1}{n}u^\top\left(1 + 1\epsilon_0\delta - D_yX\Pi_\|\theta - D_yX\Pi_\perp\theta\right) + \zeta\left(\frac{\|\Pi_\perp\theta\|_1}{\sqrt{d}} - \delta\right). \tag{40}$$

**Convex Gaussian Minimax Theorem**    Since the adversarial attacks are consistent and the observations are noiseless, we know the solution in Equation (38) exists for all $d, n$. Yet, in order to apply the CGMT, we have to show that we can restrict $u$ and $\theta$ to compact sets. This follows from a simple trick as explained in Section D.3.1 in [24]. Hence, the primal optimization problem from Equation (40) can be asymptotically replaced with the following auxiliary optimization problem, where, as before, $g \in \mathbb{R}^d$ and $h \in \mathbb{R}^n$ are random vectors with i.i.d. standard normal entries:

$$\min_{\theta,\delta} \max_{\substack{u:u_i\geq 0,\\ \zeta\geq 0}} \|\theta\|_2^2 + \frac{1}{n}u^\top\left(1 + 1\epsilon_0\delta - D_yX\Pi_\|\theta + D_yh\|\Pi_\perp\theta\|_2\right)$$
$$+ \frac{1}{n}\|u\|_2 g^\top\Pi_\perp\theta + \zeta\left(\frac{\|\Pi_\perp\theta\|_1}{\sqrt{d}} - \delta\right). \tag{41}$$

**Scalarization of the optimization problem**    The goal is again to scalarize the optimization problem. As a first step, we can solve the optimization over $u$ when defining $r = \frac{\|u\|_2}{\sqrt{n}}$:

$$\min_{\theta,\delta\geq 0} \max_{\substack{r\geq 0,\\ \zeta\geq 0}} \|\theta\|_2^2 + \frac{r}{\sqrt{n}}\|\max\left(0, 1 + 1\epsilon_0\delta - D_yX\Pi_\|\theta + D_yh\|\Pi_\perp\theta\|_2\right)\|_2$$
$$+ \frac{r\sqrt{\gamma}}{\sqrt{d}}g^\top\Pi_\perp\theta + \zeta\left(\frac{1}{\sqrt{d}}\|\Pi_\perp\theta\|_1 - \delta\right),$$

where $\max$ applies element-wise over the vector. We can now swap maximization and minimization since the objective is convex in $\theta, \delta$ and concave in $r$:

$$\max_{\substack{r\geq 0,\\ \zeta\geq 0}} \min_{\theta,\delta\geq 0} \|\theta\|_2^2 + \frac{r}{\sqrt{n}}\|\max\left(0, 1 + 1\epsilon_0\delta - D_yX\Pi_\|\theta + D_yh\|\Pi_\perp\theta\|_2\right)\|_2$$
$$+ \frac{r\sqrt{\gamma}}{\sqrt{d}}g^\top\Pi_\perp\theta + \zeta\left(\frac{1}{\sqrt{d}}\|\Pi_\perp\theta\|_1 - \delta\right).$$

We now want to separate $\|\Pi_\perp\theta\|_2$ from the term in $\max$. This is achieved by introducing the variable $\nu_\perp \geq 0$ and the Lagrange multiplier $\kappa$. Further, we set $\nu_\| = \langle \theta^\star, \Pi_\|\theta\rangle$ (recall that $\theta^\star = (1, 0, \cdots, 0)$), which allows us to equivalently write

$$\max_{\substack{r\geq 0,\\ \zeta\geq 0}} \min_{\substack{\nu_\perp\geq 0,\\ \nu_\|,\delta\geq 0,\Pi_\perp\theta}} \max_{\kappa\geq 0} \nu_\|^2 + \|\Pi_\perp\theta\|_2^2 + \kappa(\|\Pi_\perp\theta\|_2 - \nu_\perp)$$
$$+ \frac{r}{\sqrt{n}}\|\max\left(0, 1 + 1\epsilon_0\delta - D_yX\Pi_\|\theta^*\nu_\| + D_yh\nu_\perp\right)\|_2 \tag{42}$$
$$+ \frac{r\sqrt{\gamma}}{\sqrt{d}}g^\top\Pi_\perp\theta + \zeta\left(\frac{1}{\sqrt{d}}\|\Pi_\perp\theta\|_1 - \delta\right).$$

Next, we use again the trick $\min_{\tau\geq 0}\frac{x^2}{2\tau} + \frac{\tau}{2} = x$, which yields

$$\max_{\substack{r\geq 0,\\ \zeta\geq 0}} \min_{\substack{\nu_\perp\geq 0,\\ \nu_\|,\delta\geq 0,\Pi_\perp\theta}} \max_{\kappa\geq 0}\min_{\tau\geq 0} \nu_\|^2 + \|\Pi_\perp\theta\|_2^2 - \kappa\nu_\perp + \frac{\kappa}{2\tau}\|\Pi_\perp\theta\|_2^2$$
$$+ \frac{\kappa\tau}{2} + \frac{r}{\sqrt{n}}\|\max\left(0, 1 + 1\epsilon_0\delta - D_yX\Pi_\|\theta^*\nu_\| + D_yh\nu_\perp\right)\|_2 \tag{43}$$
$$+ \frac{r\sqrt{\gamma}}{\sqrt{d}}g^\top\Pi_\perp\theta + \zeta\left(\frac{1}{\sqrt{d}}\|\Pi_\perp\theta\|_1 - \delta\right).$$

In a next step, note that due to high dimensional concentration, we have that

$$\frac{r}{\sqrt{n}}\| \max\left(0, 1 + 1\epsilon_0\delta - D_y X\Pi_\| \theta^* \nu_\| + D_y h\nu_\perp\right)\|_2$$

$$\overset{\text{LLN}}{\to} r\sqrt{\mathbb{E}_{Z_\|, Z_\perp}\left[\max\left(0, 1 + \epsilon_0\delta - |Z_\|| \nu_\| + Z_\perp\nu_\perp\right)^2\right]} =: \sqrt{T},$$

where $Z_\|, Z_\perp$ are standard Gaussian distributed random variable. Next, by completion of the squares we get

$$\max_{\substack{r\geq 0, \\ \zeta\geq 0}} \min_{\substack{\nu_\perp\geq 0, \\ \nu_\|, \delta\geq 0, \Pi_\perp\theta}} \max_{\kappa\geq 0} \min_{\tau\geq 0} \nu_\|^2 + (1 + \frac{\kappa}{2\tau})\|\Pi_\perp\theta + \frac{r\sqrt{\gamma}}{\sqrt{d}2(1 + \frac{\kappa}{2\tau})}g\|_2^2 - \frac{r^2\gamma}{4(1 + \frac{\kappa}{2\tau})}\|g/\sqrt{d}\|_2^2$$

$$- \kappa\nu_\perp + \sqrt{T} + \zeta\left(\frac{1}{\sqrt{d}}\|\Pi_\perp\theta\|_1 - \delta\right) + \frac{\kappa\tau}{2}$$

with $\|g/\sqrt{d}\|_2^2 \to 1$. Next, note that we can again swap minimization and maximization due to the convexity and concavity, respectively, of the optimization. Hence, we can rewrite

$$\max_{\substack{r\geq 0, \\ \zeta\geq 0}} \min_{\substack{\nu_\perp\geq 0, \\ \nu_\|, \delta\geq 0}} \max_{\kappa\geq 0} \min_{\tau\geq 0, \Pi_\perp\theta} \nu_\|^2 + (1 + \frac{\kappa}{2\tau})\|\Pi_\perp\theta + \frac{r\sqrt{\gamma}}{\sqrt{d}2(1 + \frac{\kappa}{2\tau})}g\|_2^2 - \frac{r^2\gamma}{4(1 + \frac{\kappa}{2\tau})}\|g/\sqrt{d}\|_2^2$$

$$- \kappa\nu_\perp + \sqrt{T} + \zeta\left(\frac{1}{\sqrt{d}}\|\Pi_\perp\theta\|_1 - \delta\right) + \frac{\kappa\tau}{2}.$$

Finally, to obtain the desired optimization problem, we only need to solve the inner optimization over $\Pi_\perp\theta$. For this, we can write:

$$\min_{\Pi_\perp\theta} (1 + \frac{\kappa}{2\tau})\|\Pi_\perp\theta + \frac{r\sqrt{\gamma}}{\sqrt{d}2(1 + \frac{\kappa}{2\tau})}g\|_2^2 + \zeta\frac{\|\Pi_\perp\theta\|_1}{\sqrt{d}}$$

$$\overset{\tilde{\theta}_\perp = \frac{\Pi_\perp\theta}{\sqrt{d}}}{=} \min_{\tilde{\theta}_\perp} \frac{1}{d}(1 + \frac{\kappa}{2\tau})\|\tilde{\theta}_\perp + \frac{r\sqrt{\gamma}}{2(1 + \frac{\kappa}{2\tau})}g\|_2^2 + \zeta\frac{\|\Pi_\perp\theta\|_1}{d}$$

$$= \frac{1}{d}\sum_{i=2}^{d} \min_{(\tilde{\theta}_\perp)_i} (1 + \frac{\kappa}{2\tau})((\tilde{\theta}_\perp)_i + \frac{r\sqrt{\gamma}}{2(1 + \frac{\kappa}{2\tau})}g_i)^2 + \zeta|(\tilde{\theta}_\perp)_i|$$

$$= \frac{1}{d}2(1 + \frac{\kappa}{2\tau})\sum_{i=2}^{d} \min_{(\tilde{\theta}_\perp)_i} \frac{1}{2}((\tilde{\theta}_\perp)_i + \frac{r\sqrt{\gamma}}{2(1 + \frac{\kappa}{2\tau})}g_i)^2 + \frac{\zeta}{2(1 + \frac{\kappa}{2\tau})}|(\tilde{\theta}_\perp)_i|$$

$$= \frac{1}{d}2(1 + \frac{\kappa}{2\tau})\sum_{i=2}^{d} \ell_H(-\frac{r\sqrt{\gamma}}{2(1 + \frac{\kappa}{2\tau})}g_i, \frac{\zeta}{2(1 + \frac{\kappa}{2\tau})})$$

$$\to 2(1 + \frac{\kappa}{2\tau})\mathbb{E}_Z \ell_H\left(\frac{r\sqrt{\gamma}}{2(1 + \frac{\kappa}{2\tau})}Z, \frac{\zeta}{2(1 + \frac{\kappa}{2\tau})}\right)$$

where we solve the optimization in the fourth line with $\ell_H$ being the Huber loss, given by $\ell_H(x, y) = \begin{cases} 0.5x^2 & |x| \leq y \\ y(|x| - 0.5y) \end{cases}$. Finally, we can conclude the proof from

$$\mathbb{E}_Z \ell_H(aZ, b) = \frac{a^2 + b^2}{2}\text{erf}\left(\frac{b}{\sqrt{2}a}\right) - \frac{b^2}{2} + \frac{ab}{\sqrt{2\pi}}\exp\left(-\frac{b^2}{2a^2}\right).$$

**Convergence** One can check that the optimization problems defined in Equations (26),(40) are convex in the variables which we minimize over and concave in the variables which we maximize over. Indeed, Equation (40) is immediate and Equation (26) follows straightforwardly, like before, from the fact that the desired convexity and concavity are satisfied for the problem defined in Equation (43). Thus, both problems defined in Equations (26),(40) have unique solutions. We note again that the optimum $\delta^\star$ in Equation (39) satisfies $\delta^\star = \frac{1}{\sqrt{d}}\|\Pi_\perp\theta\|$, and similarly the optima $\nu_\perp^\star$ and $\nu_\|^\star$ in Equation (43) satisfy $\nu_\perp^\star = \|\Pi_\perp\theta\|_2$ and $\nu_\|^\star = \langle\theta, \theta^\star\rangle$. Hence we can conclude the proof as the solutions of problems (26), (40) concentrate asymptotically as $d, n \to \infty$ around the same optima.