# OpenReview forum: "Interpolation can hurt robust generalization even when there is no noise"
_NeurIPS.cc/2021/Conference — NeurIPS 2021 Poster_

### Official Review · Reviewer_g2yR · 2021-07-01

**Rating:** 7
**Confidence:** 3

**Summary:**

The authors analyze the generalization performance of regularized adversarial linear regression and classification models for standard Gaussian data in the overparameterized setting.  The analysis allows for l^2 weight decay as well as l^\infty perturbation of the inputs.  Using similar methods to Hastie, Montanari, Rosset, and Tibshirani [18] and Javanmard and Soltanolkotabi [19], they are able to develop precise formula for the generalization error in the limit as n/d -> constant as n,d -> infinity.  They demonstrate that non-zero regularization has strictly better generalization in the robust setting.


**Limitations And Societal Impact:**

Yes, they adequately addressed the limitations.

**Main Review:**

At a high-level, the technical tools seem to be straightforward applications of theory developed in [18] (for Theorem 3.1) and in the CGMT/[19] (for Theorem 4.1) to the robust and l2 regularization setting, with the caveat that [19] was for the mixture of Gaussians setting rather than the Gaussian marginal setting which thus requires an additional extension.  (I did not check all of the details of the proof.) However, no previous work had been able to demonstrate concretely that nonzero ridge regularization will provably lead to better generalization performance even when there are noiseless observations in the robust setting, which as an interesting observation and informs our understanding of interpolating models. I particularly liked the intuitive explanations given in Sections 3.4 and 4.4.


A few comments/questions:

(1) It is unclear to me from Figure 2 if either the standard interpolating or regularized (asymptotic) risk is supposed to be larger in the noiseless setting.  They clearly look the same, but are the theoretical predictions the same?  It is hard to parse given the complicated formula for P and B, and the possible dependence on gamma.  It would help if there were comments on this in the text.

(2) What is the reason for requiring epsilon = O(1/sqrt(d)) in Theorem 4.1?  Can larger perturbations be tolerated, or what is the bottleneck?  Previous work [3', Corollary 2.10] has shown that adversarially trained linear classifiers over Gaussian marginals can tolerate \ell^\infty perturbations of constant order (albeit with suboptimal generalization guarantees).

(3) Can you comment on how the results would extend to l^2 perturbations?  It seems like most of the analysis would follow through, but I don't recall a discussion about this in the text.

(4) A discussion on a number of references [1', 2', 3'] would be helpful, especially [2'] as it directly concerns the ability of ridge regularized linear regression to generalize.  [1', 3'] concern the ability of adversarially trained linear classifiers (under logistic loss) to have small robust risk and/or the implicit bias of such algorithms; discussing these works would improve the presentation around lines 221-230 and 633-661.  Especially in footnote 2, the work [1'] is relevant.

(5) Relatedly, it is not clear to me if the comment on lines 221-223 is true.  Has it been shown that the gradient descent on the robust empirical risk converges to the robust maximum margin direction for the l^\infty perturbation?  [1'] shows this is the case for l^2 perturbations but I am not aware of a work that shows it for l^\infty, but perhaps the authors can point out such a work or my misunderstanding.

(6) The current main section of the text has very limited description of the proof the proof techniques involved for the results, especially Theorem 3.1 (CGMT tools seem the workhorse for Thm 4.1).  It would be helpful if more high-level proof ideas were provided in the main part of the text.

Minor points/typos:
* Figure 4 caption: "plottet"
* Figure 5 caption: "average margin (horizontal lines) decreases" should be increases, no?
* Figure 5 caption: "trendis"
* lines 303-304 appears a sentence got repeated
* lines 666-668 check writing, appears something went wrong
* lines 829-831 put () around eqs
* display (33) "sucht"


[1'] Implicit Bias of Gradient Descent based Adversarial Training on Separable Data. Yan Li, Ethan X.Fang, Huan Xu, Tuo Zhao. ICLR 2021.

[2'] Benign overfitting in ridge regression. A. Tsigler, P. L. Bartlett. 2020 preprint

[3'] Provable Robustness of Adversarial Training for Learning Halfspaces with Noise. Difan Zou, Spencer Frei, Quanquan Gu.  ICML 2021.


** post rebuttal **

Thanks for the detailed response.  I'm happy with the paper and think they did a good job of responding to other reviewers' concerns.  I recommend acceptance for the paper.


**Time Spent Reviewing:**

6

---

> ### Author Response · Authors · 2021-08-10
> **Response to Reviewer g2yR**
>
> We thank Reviewer g2yR for appreciating the relevance of our work. Their high-level summary exactly captures the key message we want to convey with our paper. We now address the main points raised in the review separately.
>
> **Response to 1): Optimal $\lambda$ for standard risk in noiseless linear regression.** As correctly noted by the reviewer, the optimal $\lambda_\text{opt}$ is zero for the standard risk in the noiseless case. This is indeed predicted by our theory by plugging $\epsilon = 0$ and $\sigma=0$ into Theorem 3.1. We added a comment on this in a revised version of our paper.
>
> **Response to 2)(i): Reason for picking $\epsilon = O(1/\sqrt(d))$ vs. larger $\epsilon$ for classification.**  We would like to thank Reviewer g2yR for raising this important question. We appreciate that we can explain this in more detail since we could not include a discussion in the main text due to space constraints.
> The primary reason for choosing $\epsilon = 1/\sqrt{d}$ is that this leads to asymptotic predictions which closely match our experimental simulations for finite $d,n$ run with small attack sizes $\epsilon$ (see Figure 4a and Appendix B for experimental details).
>
> We now further argue from a technical perspective how a vanishing $\epsilon = O(1/\sqrt{d})$ is necessary to allow non-vanishing robust margins of the estimator
> $$\min\_i \langle x\_i, \theta/\|\theta\|\_2\rangle - \epsilon \| \Pi\_{\perp} \theta/\|\theta\|\_2 \|\_1 .$$
>
> Minimizing the loss function in Equation (23) leads to a tradeoff between maximizing the margin $\min\_i y\_i\langle x\_i, \theta/\|\theta\|\_2 \rangle$ and minimizing the term  $\max\_{\| \delta\|\_{\infty} \leq \epsilon} \langle \delta, \theta/\|\theta\|\_2\rangle =  \epsilon \|\Pi\_{\perp} \theta\|\_1/ |\theta\|\_2$.  In order to have an asymptotically non-vanishing margin (and thus also robust margin),  the estimator has to align with $n \asymp d$ approximately orthogonal vectors $x_i$. Hence, the $\ell_1$-norm of the resulting estimator $\theta$ would grow when increasing $d$. In fact, you can even see that the $\ell_1$-norm of the max-$\ell_2$-margin estimator grows at rate $\sqrt{d}$ (note that this follows from Figure 4a and the definition of the robust risk).
> At the same time, however, the margin is non-zero and bounded for any $\theta$ as a consequence of the existence of a finite solution of the optimization problem in Theorem F.2. Finally, by choosing $\epsilon = O(1/\sqrt{d})$, we have $\epsilon \| \Pi_{\perp}\theta/\|\theta\|_2\|_1  = O(1)$ and hence both terms are of the same order.
>
> **Response to 2)(ii): Effect of large $\epsilon$.**  However, the question of what happens when $\epsilon$ is constant as $d \to \infty$ is still interesting. In the case of sparse ground truths, not only can they be “tolerated” as shown in [3], but in fact we observe in Figure 7 that larger $\epsilon$ in the noiseless case may even lead to consistent estimation! This is in fact the second reason why we did not discuss this setting as it is another one of the examples for when ridge regularization does not benefit regularization.
>
> We want to briefly give some intuition for why this might be the case by referring to the expression of the robust loss derived in Equation (23). It is given by an average of $\ell(y\_i \langle x\_i, \theta \rangle - \epsilon \| \Pi\_{\perp} \theta\|\_1)$, and hence varying $\epsilon$ leads to a  tradeoff between maximizing the margins $y\_i \langle x\_i, \theta \rangle$ and minimizing the projected $\ell\_1$-norm $\| \Pi\_{\perp} \theta\|\_1$.
> For large perturbations $\epsilon$ in particular, although counterintuitive, we can recover the sparse ground truth since the $\ell\_1$-penalty induces a sparsity bias that aligns with the ground truth. Since the robust max-$\ell\_2$-margin interpolator is already consistent (i.e., achieves vanishing risks as $d,n \to \infty$), the benefits of ridge regularization are limited.
>
> Finally, we would like to mention that, as a future work, a thorough non-asymptotic analysis that mathematically captures this intuition would greatly improve our understanding of robust overfitting in noiseless settings.
>
>
> **Response to 3): $\ell_\infty$ vs. $\ell_2$-perturbations in classification.** Since this is a natural question to ask, we did already investigate $\ell_2$-perturbations for robust logistic regression experimentally and think about them theoretically. However, ridge regularization does not benefit robustness in that scenario: $\ell_2$-perturbations add a $\ell_2$-penalty to the logistic loss, which leads to a shrinkage of the $\ell_2$-norm of the estimator $\theta$.
>
>
>
>
> In particular, this results in a similar effect to adding an explicit ridge ($\ell_2$) penalty, and since we do not observe any robust overfitting for standard training (see Figure 10), we also do not expect any benefits of ridge regularization when training with $\ell_2$-perturbations. Our intuition was confirmed by experiments that we can add in a revised version together with a discussion.
>
> Lastly, the proof of Theorem 4.1 can be extended to $\ell_2$-perturbations straightforwardly, but requires constant $\epsilon$ in order to observe any effect of adversarial training. Note that this stands in contrast to the case of $\ell_{\infty}$-perturbations as discussed in the response to question 2). Essentially, the main reason is that in the case of $\ell_2$-perturbations, $\max_{\| \delta\|_{2} \leq \epsilon} \langle \delta, \theta/\|\theta\|_2\rangle = \epsilon$ and thus the influence of the adversarial perturbations is bounded even for constant attack sizes $\epsilon$ as $d\to \infty$.
>
> To give a more detailed answer regarding how to modify the proof  in the case of $\ell\_2$-perturbations: Essentially, we would only have to replace $\|w\|\_{\infty}$ with $\|w\|\_2$ on line 809 in the proof of Theorem F.1. For the choice $\epsilon = O(1/\sqrt{d})$, the optimal $w \approx 0$, which means that adversarial attacks have asymptotically no influence. On the other hand, for a constant $\epsilon$, the optimal $\|w\|\_2$ is non-zero. In particular, in this case, a modification of the section on lines 819-822  captures the tradeoff between minimizing the adversarial penalty (which has the closed form expression $\epsilon \|\Pi\_{\perp} \theta\|\_2$ when training with $\ell\_2$-perturbations) and maximizing the margins $y\_i \langle x\_i, \theta \rangle$. Finally, a similar argument also holds for the proof of Theorem F.2.
>
>
> **Response to 4): Discussion on additional references.** We would like to thank the reviewer for pointing out these references. Reference [1] is a more accurate citation than [4] for the convergence of the gradient descent path of robust logistic regression. Furthermore, reference [3] studies robust estimators trained with gradient descent and shows that early stopping can lead to robustness to noise. This is a relevant citation which we added to the discussion of Figure 8, emphasizing how early stopping prevents the robust logistic regression estimate from overfitting.
> Lastly, we agree that  [2] is another important contribution providing tight bounds for the standard risk of linear ridge regression and min-$\ell_2$-norm interpolators for overparameterized regression. Our focus, however, is on understanding the heavily overparameterized regime. There, previous works [4,5] show that ridge regularization is not beneficial compared to min-$\ell_2$-norm interpolation. We would like to refer to the section in our general response highlighting our main contribution and how it differs from other works that study regularized estimators.
>
>
> **Response to 5): Convergence of gradient descent to robust max-margin.** In our paper we primarily cite Corollary 3.2 of [6] where the authors point out that a straightforward consequence of [7,8] is that gradient descent minimizing the unregularized robust logistic loss in Equation (8) converges to the robust max-$\ell_2$-margin interpolator from Equation (9). In fact, reference [1] pointed out by the reviewer explicitly proves the fact that gradient descent converges to the robust max-$\ell_2$-margin via their Definition 3.2, Lemma 3.2 and Theorem 3.4. Since [1] provides a much cleaner and more rigorous statement, we changed the citation in our revised version of the paper.
>
> **Response to 6): Details on the proofs of the theorems.** We addressed this shortcoming by adding a comment on the methodologies used in Theorem 3.1 to the revised version of our paper.
>
> **Typos.** We would like to thank Reviewer g2yR for pointing out the typos. We fixed them in our revised manuscript.
>
> [1] Implicit Bias of Gradient Descent based Adversarial Training on Separable Data. Yan Li, Ethan X.Fang, Huan Xu, Tuo Zhao. ICLR 21.\
> [2] Benign overfitting in ridge regression. A. Tsigler, P. L. Bartlett. Arxiv 20.\
> [3] Provable Robustness of Adversarial Training for Learning Halfspaces with Noise. Difan Zou, Spencer Frei, Quanquan Gu. ICML 21.\
> [4]: Optimal Regularization Can Mitigate Double Descent. Preetum Nakkiran, Prayaag Venkat, Sham Kakade, Tengyu Ma. ICLR 21.\
> [5]: Surprises in High-Dimensional Ridgeless Least Squares Interpolation. Trevor Hastie, Andrea Montanari, Saharon Rosset, Ryan J. Tibshirani. Arxiv 18. \
> [6]: Precise Statistical Analysis of Classification Accuracies for Adversarial Training. Adel Javanmard and Mahdi Soltanolkotabi. Arxiv 20. \
> [7]: Risk and parameter convergence of logistic regression. Z. Ji and M. Telgarsky. COLT 19.\
> [8]: The implicit bias of gradient descent on separable data. D. Soudry, E. Hoffer, M. S. Nacson, S. Gunasekar, and N. Srebro. JMLR 18.

---

### Official Review · Reviewer_yfXT · 2021-07-06

**Rating:** 5
**Confidence:** 3

**Summary:**

The paper studies the benefits of non-vanishing regularization for improving generalization in overparameterized classification and regression problems. In this setting, previous works studied the implicit bias effect. Particularly, it was shown that training models *without* explicit regularization can still yield good generalization performance. In contrast, in this paper, the authors show that explicit regularization can further improve generalization, especially w.r.t. the adversarial robust risk.

**Ethical Concerns:**

None.

**Limitations And Societal Impact:**

It is hard to properly assess the limitations of the results due to the reasons listed above.

**Main Review:**

The paper studies how regularization affects the adversarial robust risk for both linear and logistic regression problems.

The paper main contributions are:
-	The authors illustrate empirically the benefits of ridge regression and early stopping w.r.t. robust generalization, both for linear and logistic regression.
-	The authors state theoretical results which shed light on these empirical observations.

My main concern is the paper’s writing and clarity. I found the flow of the paper lacking and, in general, the message the authors tried to convey did not pass clearly. In addition, there are many sloppy typos throughout the draft. Doing a very thorough revision and improving the writing flow would greatly benefit the paper and make the message much clearer.

Some examples of issues that should be addressed:
-	Theorem 3.1 seems to be cut off in the middle of a sentence. The Theorem abruptly ends with the sentence “Finally,”.
-	Lines 303-305: “While our theoretical results hold for the ridge penalty, we experimentally observe the same behavior for early stopping gradient descent. While our theoretical results apply for ridge regularization, we also observe the same behavior for early stopping.”
This seems to be just a duplication of the same sentence and is probably a typo.
-	The observations and conclusions from each of the experiments should be clearly stated.
-	The Theorems are very technical, and some explanations will help better understand the results. For example, how is the function m behaves as a function of z? What this means on B and V depends on $\lambda$? There is some discussion in the paper, but I found it insufficient.

Items such as these persist throughout the paper and make it difficult to assess the paper quality, correctness, and the significance of the results.

---

After reading the rebuttal: In their response, the authors made the paper's main contributions much clearer. Therefore, I would like to increase the score to 5. I hope that, given the reviewers’ feedback, the authors will improve the clarity of the final draft.

**Time Spent Reviewing:**

8

---

> ### Author Response · Authors · 2021-08-10
> **Response to Reviewer yfXT**
>
>
> We thank Reviewer yfXT for the feedback on our work. We now address the main points raised in the review.
>
> > My main concern is the paper’s writing and clarity. I found the flow of the paper lacking and, in general, the message the authors tried to convey did not pass clearly. In addition, there are many sloppy typos throughout the draft. Doing a very thorough revision and improving the writing flow would greatly benefit the paper and make the message much clearer.
>
> **Typos and clarity of writing**: We have made substantial improvements in the exposition of the paper, detailed in the general comments, which as we hope conveys our message much clearer.
>
>
> > The observations and conclusions from each of the experiments should be clearly stated.
>
> **Discussion of the experiments**: The submitted paper states the main take-aways for each experiment in the respective figure caption. In addition to those, we provide a detailed interpretation of each result in the main text. A detailed description of the experimental setting is in the supplementary material. We kindly ask Reviewer yfXT to point out which experiments were unclear in particular, so that we can point to the respective explanation, improve those explanations, and/or provide clarifying remarks.
>
>
> > The Theorems are very technical, and some explanations will help better understand the results. For example, how is the function m behaves as a function of z? What this means on B and V depends on $\lambda$? There is some discussion in the paper, but I found it insufficient.
>
> **Intuition for theorems**: While both theorems are rather technical, we additionally provide intuition that can help to understand their significance. For example, we plot the asymptotic values for the standard and the robust risks from Theorems 3.1 and 4.1 in Figures 2 and 4 respectively. In both cases we show that regularization helps to achieve a lower asymptotic robust risk and that the trend is closely matched by simulations for finite $d$ and $n$. Furthermore, in Sections 3.4 and 4.4, we conduct additional experiments in order to gain further insights on the phenomena causing robust overfitting for linear and logistic regression (see Figures 3 and 5, respectively).
>
> **Interpretation of the quantities $B,V,m(z)$ that appear in Theorem 3.1**: As we mention in the main text, Theorem 3.1 is an extension for the robust risk of a result originally derived by Hastie et al. for the standard risk. Therefore, many of the quantities in Theorem 3.1 have been introduced in [1]. In particular, $B$ and $V$ represent the asymptotic bias and variance, respectively, as we mention in the theorem statement. Furthermore, the function $m(z)$ is the Stieltjes transform of the Marchenko-Pastur law. We added a comment on this in a revised version of our paper.
>
> [1]: Surprises in High-Dimensional Ridgeless Least Squares Interpolation. Trevor Hastie, Andrea Montanari, Saharon Rosset, Ryan J. Tibshirani. Arxiv 18.

---

### Official Review · Reviewer_U2TN · 2021-07-17

**Rating:** 7
**Confidence:** 2

**Summary:**

This paper studies how regularization affects the robust population risk in the overparameterization regime. The authors study the behavior of linear regression and logistic regression under sufficient statistical settings. In particular, even in the noiseless regime, the population robust risk benefits from the regularization, but not the population test risk. The argument is empirically verified.

**Limitations And Societal Impact:**

Yes.

**Main Review:**

The problem studied is important and interesting. The writing is clear and the proofs look technically sound. These results help us further understand the behavior of overparameterization, especially in the adversarial environment. Here are my questions and some minor typos:

1. In theorem 3.1 (and also 4.1), how do we get $R(\hat\theta_0)$ or $R_\epsilon(\hat\theta_0)$? Do we directly take the limit of the RHS in (6) as $\lambda\to 0$? If so there should be an argument that involves monotone convergence/dominated convergence theorem that exchanges the limit and integration (which may be straightforward but worth mentioning)? Say I set $\gamma=2$, then isn't $\lim_{\lambda\to 0^+}m(-\lambda)=\infty$? I probably missed something here.
2. In Figure 4(b), when $1/\lambda\to 0$, the ERM should start to look like an all $0$ vector right? Then shouldn't the risk be large?
3. Is there any intuition that you use $\ell_2$ regularization other than $\ell_1$ in (8) since the true $\theta^*$ is assumed to be sparse? I guess one thing is $\ell_2$ is easier to analyze, but do we at least know how $\ell_1$ regularization behaves in this setting?
4. [1] is an important reference that is missing. They also study the relation between regularization and double descent, but in the non-adversarial setting.
5. In the description of figure 5 some typos: "the minimum margin is decreases...", "this trendis also observed..."
6. Line 257, "regularization improves regularization".

[1]: Optimal Regularization Can Mitigate Double Descent

**Time Spent Reviewing:**

4

---

> ### Author Response · Authors · 2021-08-10
> **Response to Reviewer U2TN**
>
>
> We thank Reviewer U2TN for appreciating the relevance of our work. We now address the main points of the review separately.
>
>
> **Response to 1): Limit of $\lambda \to 0$ for min-norm interpolator results in Theorem 3.1.** We obtain the asymptotic risk of the min-$\ell_2$-norm interplator by taking the limit $\lim_{\lambda \to 0}$ of the RHS in Equation (6). We remark that $m(z)$ is the Stieltjes transform of the Marchenko-Pastur law. Hence, if $\gamma \neq 1$ then the limit $\lim_{z \to 0} m(z)$ exists and is $\lim_{z \to 0} m(z) = 1/(1-\gamma)$ if $\gamma <1$ and $\lim_{z \to 0} m(z) = 1/(\gamma(\gamma -1 ))$ if $\gamma >1$; not infinity as indicated by Reviewer U2TN. For further details we would like to point the reader to Corollary 5 in [2]. We added the above remarks to  Theorem 3.1 and its proof in the revised version of our paper.
>
>
> **Response to 2): Large $\lambda$ in robust logistic ridge regression inducing zero vector and large risk.** We point out that for *classification* the risk defined with the 0-1 loss only depends on the direction of the estimator, but not on the norm. Therefore, a large $\lambda$ does not contradict the good performance the estimator attains in Figure 4b for large $\lambda$.
>
>
> **Response to 3) (i): Why we study regularization with $\ell_2$-penalty.** We thank Reviewer U2TN for this question as it made us realize that we have to emphasize the motivation for ridge regularization much more clearly in the revised version of our paper.
>
> Taking the limit $\lim_{\lambda \to 0}$ of ridge ($\ell_2$) regularization results in the robust max-$\ell_2$-margin and min-$\ell_2$-norm interpolators for logistic and linear regression, respectively. The min-$\ell_2$-norm interpolator is also the interpolating estimator obtained from *unregularized gradient descent*.
> While this fact is well-known, we also empirically observe that the optimization path of robust logistic regression with decreasing ridge penalty and gradient descent exhibit similar risk curves, i.e., early stopping and ridge regularization yield solutions with similar robust risks. In fact, Rice et al. [3] observe exactly that early stopping benefits robust generalization.
>
> To better highlight this analogy, we moved the discussion on the similarities between gradient descent and ridge regularization to the main text in the revised version of our paper. In particular, we encourage future work to study the gradient descent path with respect to the robust risk and to provide theoretical evidence for the similarities observed in Figure 8.
>
> **Response to 3) (ii): Effect of regularization with $\ell_1$-penalty.** Nonetheless, it is perfectly reasonable to wonder what would happen if we regularized with an $\ell_1$-penalty and compared $\lambda > 0$ with $\lambda \to 0$. We briefly investigated this setting as well and came to the following conclusions:
>
> An explicit $\ell_1$-penalty induces a strong bias towards a sparse solution. Such estimators have been studied for standard linear classification and can even reach consistency when taking the limit $\lambda \to 0$ (see [4] and references therein).  In fact, we observe in our own experiments that using $\ell_1$-regularization instead of $\ell_2$ allows the estimator to achieve a vanishing robust risk, assuming that the ground truth function is sparse.
>
> Furthermore, an important contribution of our paper is that ridge ($\ell_2$) regularization can lead to unexpected benefits even in completely noiseless settings, indicating that its role goes beyond its classical perception as a variance-reduction technique. In contrast, $\ell_1$-regularization induces an implicit bias towards sparse solutions and therefore results in a very different effect. Having said that, a detailed study of the role of $\ell_1$-regularization to improve the robust risk would be an interesting direction for future work.
>
>
> **Response to 4): Suggestions for additional references.** We would like to thank Reviewer U2TN for noting that regularization in linear regression mitigates the peak of the standard risk at the interpolation threshold by reducing variance in the presence of noise, as established in [1,2,5]. In fact, it is well known that ridge regularization leads to a bias-variance tradeoff and it is therefore not surprising that explicit ridge regularization helps to mitigate the peak in the double descent curve, that is caused by the large variance. In contrast, as we already highlight in the paragraph above, in our paper we show the benefits of ridge regularization for the robust risk in
>
> - highly overparameterized settings where the effect of explicit regularization is negligible for the standard risk as also shown in [1,2]
> - noiseless settings where the variance is zero and thus the standard risk does not benefit from regularization, even at the interpolation threshold.
>
>
>
> **Response to 5,6): Typos.** We would like to thank Reviewer U2TN for pointing out the typos. We fixed them in the revised manuscript.
>
> [1]: Optimal Regularization Can Mitigate Double Descent. Preetum Nakkiran, Prayaag Venkat, Sham Kakade, Tengyu Ma. ICLR 21.\
> [2]: Surprises in High-Dimensional Ridgeless Least Squares Interpolation. Trevor Hastie, Andrea Montanari, Saharon Rosset, Ryan J. Tibshirani. Arxiv 18. \
> [3]: Overfitting in adversarially robust deep learning. Leslie Rice, Eric Wong, J. Zico Kolter. ICML 20. \
> [4]: AdaBoost and robust one-bit compressed sensing. Geoffrey Chinot, Felix Kuchelmeister, Matthias Löffler, Sara van de Geer. Arxiv 21.\
> [5] Benign overfitting in ridge regression. A. Tsigler, P. L. Bartlett. Arxiv 20.

---

> > ### Author Response · Authors · 2021-09-01
> > **Clarity of the answer**
> >
> > Since the discussion period ends soon and we would like to be able to respond to any remaining concern Reviewer U2TN may have, we would like to kindly ask Reviewer U2TN to let us know if there is anything we could further clarify.

---

> > > ### Comment · Reviewer_U2TN · 2021-09-01
> > > **Concerns addressed**
> > >
> > > Thanks for the response of the authors. My concerns are addressed and I raise my score to 7.

---

### Official Review · Reviewer_kE4t · 2021-07-18

**Rating:** 6
**Confidence:** 3

**Summary:**

This paper compares the effect of interpolation and regularization on robust generalization. It shows that in the case of regression under squared loss and classification under the log loss when the ground truth is a linear model robust generalization error is smaller for regularized estimators. The robust generalization error is defined as the error under consistent adversarial perturbations.

**Limitations And Societal Impact:**

Some limitations were addressed by the authors.

**Main Review:**

This paper provides important insight into the effect of regularization vs interpolation for robust generalization. It is clearly written and well-organized.

While the results presented in this paper are useful and interesting I have some concerns about the form of adversarial perturbations considered. In particular, there is not sufficient motivation why the particular definition of adversarial perturbation is used. It seems as though considering perturbation orthogonal to the ground truth may primarily be well suited to the linear ground truth setting. It's not clear if this is a valid choice for other models or how to generalize such perturbation to different models especially one where the data is lower dimensional. It is also unclear why "adversarial" perturbations are consistent. That is a strong assumption.

On a different axis, it is not clear why training in the regression setting does not include adversarial examples but training in the classification setting does. What structural differences in the two problems imply the use of these different settings?

Furthermore, to provide a faithful comparison to related work some experimental evidence on real world data might be beneficial.

**Time Spent Reviewing:**

3 hours

---

> ### Author Response · Authors · 2021-08-10
> **Response to Reviewer kE4t**
>
>
>
> We thank Reviewer kE4t for appreciating the relevance of our work. We now address the main points raised in the review.
>
> >[...] to provide a faithful comparison to related work some experimental evidence on real world data might be beneficial:
>
> **Prior works already provide experimental evidence on real world data.** Our work draws on two directions of related work: 1) the theoretical results that characterize the inductive bias of min-$\ell_2$-norm interpolators (for regression) and max-$\ell_2$-margin interpolators (for classification) in the overparameterized regime; and 2) the empirical observation of the phenomenon of robust overfitting [1]. We assume that Reviewer kE4t refers to the latter, but we would be happy to clarify the relationship to the former line of work as well if desired.
>
> Prior work shows experimentally that deep neural networks trained on large image datasets benefit from early stopping if evaluated with the adversarially robust risk [1], or  with the worst-case risk among subpopulations [2,3]. A recent work [4] argues that overfitting of the adversarially robust risk may be due to noise in the training data. In our paper, we instead focus on showing that robust overfitting occurs even for settings in which we can reduce label noise to a minimum (see the experiments in Figure 1 for more details).
>
>
> > While the results presented in this paper are useful and interesting I have some concerns about the form of adversarial perturbations considered. In particular, there is not sufficient motivation why the particular definition of adversarial perturbation is used. It seems as though considering perturbation orthogonal to the ground truth may primarily be well suited to the linear ground truth setting. It's not clear if this is a valid choice for other models or how to generalize such perturbation to different models especially one where the data is lower dimensional. It is also unclear why "adversarial" perturbations are consistent. That is a strong assumption.
>
> **Consistent perturbations in practice and extending the definition to generic function classes.** We assume that the reviewer is challenging the fact that adversarial perturbations in practice (e.g. on images) are consistent and asks whether this assumption for theoretical analysis is reasonable. We kindly ask Reviewer kE4t to correct our presumption if they are not satisfied with our response.
>
> Adversarial perturbations were originally defined in the context of image data to be *imperceptible* to the human eye [5], in the sense that they do not change the ground truth label, or equivalently, the ground truth classifier perfectly classifies the perturbed inputs. This property of perturbations $\delta$ is what we call consistency (as has been done before in [6]), i.e., $f^\star(x+\delta) = f^\star(x)$, for a generic ground truth $f^\star$ and all inputs $x$. We note that this definition is general and can be used for arbitrary function classes. In particular, for linear models, it takes the form of the orthogonality condition $\langle \theta^\star, \delta \rangle = 0$ that we use in our derivations.
>
>
>
> > On a different axis, it is not clear why training in the regression setting does not include adversarial examples but training in the classification setting does. What structural differences in the two problems imply the use of these different settings?
>
> **Difference between adversarial training for regression and classification.** For linear regression, adversarial training either renders interpolating estimators infeasible, or requires oracle knowledge of the ground truth which leaks too much information and  allows perfect recovery of $\theta^\star$. In contrast, for linear classification, interpolation is easier to achieve – it only requires the sign of $\langle x_i, \theta \rangle$ to be the same as the label $y_i$ for all $i$. In particular, when the data is sufficiently high-dimensional, it is possible to find an interpolator of the adversarially perturbed training set. We further elaborate on the topic of consistent adversarial training in the general comments.
>
>
> [1]: Overfitting in adversarially robust deep learning.Rice, Leslie and Wong, Eric and Kolter, Zico. ICML 20. \
> [2]: Distributionally Robust Neural Networks. Shiori Sagawa and Pang Wei Koh and Tatsunori B. Hashimoto and Percy Liang. ICLR 20. \
> [3]: An Investigation of Why Overparameterization Exacerbates Spurious Correlations. Sagawa, Shiori and Raghunathan, Aditi and Koh, Pang Wei and Liang, Percy. ICML 20. \
> [4]: How Benign is Benign Overfitting? Amartya Sanyal and Puneet K. Dokania and Varun Kanade and Philip Torr. ICLR 21. \
> [5]: Explaining and Harnessing Adversarial Examples. Ian Goodfellow and Jonathon Shlens and Christian Szegedy. ICLR 15. \
> [6]: Understanding and Mitigating the Tradeoff Between Robustness and Accuracy. A. Raghunathan, S. M. Xie, F. Yang, J. Duchi, P. Liang. ICML 20

---

> > ### Comment · Reviewer_kE4t · 2021-08-20
> > **Acknowledgement of Authors' Response**
> >
> > I have read the authors' response and my concerns have been addressed.

---

### Author Response · Authors · 2021-08-10
**To all reviewers**

We would like to thank all the reviewers including the area chair for their contribution to this conference and for helping us to improve the quality of our paper. We would further like to briefly summarize the major contributions of this paper and highlight the most prominent changes made to the revised version of our paper which already address some of the concerns of the reviewers. In addition to these changes, we point out that we have substantially improved the clarity of the paper in terms of structure and language by elaborating on some key concepts (e.g. motivation for consistent/inconsistent perturbations) and removing the typos.

**Main contribution and relation to other works on overparameterized linear regression that show benefits for regularization around the interpolation threshold.** Our paper demonstrates how the robust risk of linear models benefits from explicit ridge regularization in overparameterized settings, even if the data is *entirely noiseless*. When the goal is to achieve good robust generalization, our results contradict the modern storyline that highly overparameterized models (e.g. $d/n \gg 1$ for linear models) perform best when interpolating the training data without the need for explicit regularization. To the best of our knowledge, we are the first to provide theoretical evidence that the phenomenon of robust overfitting occurs even in entirely noiseless settings.

As noted by the reviewers, a number of recent related works [1,2,3] show that regularization in linear regression mitigates the peak of the standard risk at the interpolation threshold by reducing the variance in the presence of noise. In fact, it is well known that ridge regularization leads to a bias-variance tradeoff and it is therefore not surprising that explicit ridge regularization helps to mitigate the peak in the double descent curve, that is caused by the large variance. In contrast, in our paper we show the benefits of ridge regularization for the robust risk in

- highly overparameterized settings where the effect of explicit regularization is negligible for the standard risk as also shown in [1,2].
- noiseless settings where the variance is zero and thus the standard risk does not benefit from regularization, even at the interpolation threshold.


**Motivation for studying standard training for linear regression and adversarial training for logistic regression.** In our paper we study standard trained estimators for regression and adversarially trained estimators for classification. We motivated our choice by adding a short discussion in the revised version of our paper that captures the following argument: Since the goal of this paper is to study the shortcomings of interpolating estimators compared to regularized estimators, we only analyze training algorithms that allow interpolation. For regression, inconsistent adversarial training prevents interpolation (as discussed in Appendix C). On the other hand, consistent adversarial training simply recovers the ground truth for linear regression when training with noiseless samples. In contrast, for linear classification, interpolation is easier to achieve -- it only requires the sign of $\langle x_i , \theta\rangle$ to be the same as the label $y_i$ for all $i$. In particular, when the data is sufficiently high-dimensional, it is possible to find a classifier that interpolates the adversarially perturbed training set.


**Motivation for adversarial training with consistent perturbations for logistic regression.** As mentioned before, our main contribution is to show that robust overfitting occurs even in entirely noiseless settings, where we expect the phenomenon to happen the least. However, training with inconsistent perturbations induces noise during training, even if the clean data is noiseless. To challenge the hypothesis that noise is responsible for robust overfitting, we decided to focus on training with consistent perturbations instead. Even though the procedure is not practically feasible since it requires full knowledge of the ground truth, consistent adversarial training allows for an entirely noiseless setting. Thus, we can rigorously confirm that robust overfitting occurs even in the complete absence of noise in the training data. However, based on the reviewers’ feedback, we think that it is important to also discuss inconsistent adversarial training more in detail. We have revised our paper accordingly and now present empirical results for linear regression with both consistent and inconsistent adversarial training in the main text. Moreover, we have adapted Theorem 4.1 such that it holds for both training procedures.

We motivate our choice of the training algorithm in the revised paper as follows:

> Adversarial training with respect to inconsistent perturbations is a popular choice in the literature to improve the robust risk (e.g., [4,5]).  The terminology is derived from the fact that adversarial training with such perturbations is inconsistent in the infinite data limit for fixed $d$. The failure to recover the ground truth even for noiseless data is caused by perturbed samples that may cross the true decision boundary. In other words, inconsistent perturbations effectively introduce noise during the training procedure.

> As mentioned in the introduction, in this paper we are interested in verifying whether regularization can be beneficial in high dimensions even in the absence of noise. However, adversarial training with inconsistent perturbations can induce noise even if the data is clean. In particular, in the data model with noiseless observations that we introduce in Section 2.1 **(previously Section 4.1)**, the ground truth function misclassifies approximately 8% of the labels when perturbing the training data with inconsistent perturbations of size $\epsilon = 0.1$. Therefore, in the sequel we study the impact of both inconsistent and consistent perturbations on robust overfitting.



[1]: Optimal Regularization Can Mitigate Double Descent. Preetum Nakkiran, Prayaag Venkat, Sham Kakade, Tengyu Ma. ICLR 21. \
[2]: Surprises in High-Dimensional Ridgeless Least Squares Interpolation. Trevor Hastie, Andrea Montanari, Saharon Rosset, Ryan J. Tibshirani. Arxiv 18. \
[3] Benign overfitting in ridge regression. A. Tsigler, P. L. Bartlett. Arxiv 20.  \
[4]: Explaining and Harnessing Adversarial Examples. Ian Goodfellow and Jonathon Shlens and Christian Szegedy. ICLR 15. \
[5]: Precise Statistical Analysis of Classification Accuracies for Adversarial Training. Adel Javanmard and Mahdi Soltanolkotabi. Arxiv 20.

---

### Decision · Program_Chairs · 2021-09-27

**Decision:**

Accept (Poster)

**Comment:**

This paper aims to understand the effect of ridge regularization in overparameterized settings. It is believed that overparameterized models already have implicit bias and incorporating regularization does not provide any additional benefits in terms of standard risk. However, this paper shows that, for linear models, employing ridge regularization does help improve robust generalization in both regression and classification settings, even when the training is performed with noiseless data.

All reviewers agree that the paper studies an interesting and timely problem, and makes multiple novel theoretical contributions. There were some concerns about the writing of the paper that have been resolved via the discussion among the authors and the reviewers. Once reviewers' feedback is incorporated, this paper will be a valuable addition to NeurIPS 2021.